# Mechanisms of axoneme and centriole elimination in *Naegleria gruberi*

Alexander Woglar, Coralie Busso [ID], Gabriela Garcia-Rodriguez [ID], Friso Douma [ID], Marie Croisier, Graham Knott & Pierre Gönczy [ID] [✉]

## Abstract

The early branching eukaryote *Naegleria gruberi* can transform transiently from an amoeboid life form lacking centrioles and flagella to a flagellate life form where these elements are present, followed by reversion to the amoeboid state. The mechanisms imparting elimination of axonemes and centrioles during this reversion process are not known. Here, we uncover that flagella primarily fold onto the cell surface and fuse within milliseconds with the plasma membrane. Once internalized, axonemes are severed by Spastin into similarly-sized fragments that are then enclosed by membranes, before their contents are eliminated through the lysosomal pathway. Moreover, we discovered that centrioles undergo progressive K63 autophagy-linked poly-ubiquitination and K48 proteasome-promoting poly-ubiquitination, and that such ubiquitination occurs next to centriolar microtubules. Most centrioles are eliminated in either lysosomes or the cytoplasm in a lysosomal- and proteasome-dependent manner. Strikingly, we uncover in addition that centrioles can be shed in the extracellular milieu and taken up by other cells. Collectively, these findings reveal fundamental mechanisms governing the elimination of essential cellular constituents in *Naegleria* that may operate broadly in eukaryotic systems.

**Keywords** Centriole; Organelle Homeostasis; Flagellar Axoneme; Lysosome Proteasome
**Subject Categories** Cell Adhesion, Polarity & Cytoskeleton; Microbiology, Virology & Host Pathogen Interaction

## Introduction

Flagella and centrioles play indispensable roles across eukaryotic systems. Flagella, which are also known as motile cilia, are instrumental for cell motility, whereas centrioles are crucial organizers of the microtubule cytoskeleton, being pivotal notably for cell division in animal cells. Centrioles exhibit a near-universal 9-fold radially symmetrical arrangement of microtubule blades that is imparted onto the axoneme of the flagella they template. Whereas there has been substantial progress in recent years in understanding how flagella and centrioles assemble, knowledge regarding the mechanisms governing their elimination is limited in comparison.

The removal of flagella takes place in various physiological settings and can occur through shedding into the external milieu, shortening, or internalization into the cell body (reviewed in Bloodgood, 1974). Shedding, also referred to as deflagellation, occurs through membrane fission and axoneme severing at the base of the flagellum, and is typically triggered by an influx of calcium (reviewed in Mirvis et al, 2018). Progressive shortening of the flagellum occurs for instance in *Chlamydomonas reinhardtii*, catalyzed by the microtubule depolymerizing enzyme Kinesin-13 (Piao et al, 2009). Whereas the microtubule-severing enzyme Katanin has been proposed to play a role as well by enabling polymer cutting at the proximal end of the flagellum (Lohret et al, 1998; Rasi et al, 2009), deflagellation occurs normally in a katanin p60 null mutant strain (Dymek and Smith, 2012). Axonemal internalization into the cell body has been observed in many unicellular organisms, and can occur through distinct modes (reviewed in Bloodgood, 1974). First, the axoneme, but not the surrounding membrane, can be retracted into the cytoplasm, as observed for example in the oral apparatus of *Tetrahymena pyriformis* (Moore, 1972; Williams and Frankel, 1973) or the zygote of *Ulva mutabilis* (Braten, 1971). Second, the flagellum can contact the cell body laterally, after which the flagellar membrane is hypothesized to fuse laterally with the plasma membrane, resulting in axoneme internalization. This is the case for instance also in *Ulva mutabilis* (Braten, 1971), as well as in chytrid fungi (Venard et al, 2020). In a third mode of internalization observed for example in *Allomyces arbuscula* (Barrett, 1912; Ritchie, 1947) and *Siphonaria variabilis* (Sparrow, 1937), the flagellar membrane forms a vesicle into which the axoneme rolls up and which then fuses with the plasma membrane, resulting in axoneme internalization. In the few systems where this question has been studied (e.g., Moore, 1972; Venard et al, 2020), the axoneme is degraded following internalization through mechanisms that remain to be fully understood, although it is clear that the proteasome contributes to this process in chytrid fungi (Venard et al, 2020). *Trypanosoma cruzi* follows yet another route to produce cells without cilia: when invading the host, cells undergo an asymmetric division that produces one daughter cell with a nucleus but no flagellum, and a

Swiss Institute for Experimental Cancer Research (ISREC), School of Life Sciences, Swiss Federal Institute of Technology Lausanne (EPFL), Lausanne, Switzerland.
✉E-mail: pierre.gonczy@epfl.ch

second daughter cell with a flagellum but no nucleus which degenerates shortly after division (Kurup and Tarleton, 2014).

Another setting in which the flagellar axoneme is removed occurs after fertilization in metazoan organisms. In some systems, including mouse embryos, the flagellar axoneme is dismantled rapidly in the zygote after fertilization, such that merely axonemal fragments are present by the time of pronuclear fusion (Thompson et al, 1974). In bovine embryos, the proteasome localizes to the internalized axoneme, such that the degradation complex might participate in axoneme elimination (Sutovsky, 2018), as in chytrids (Venard et al, 2020). Conceivably, degradation of the sperm tail axoneme in the zygote could be required to ensure that mitochondria are only contributed maternally and not from the paternal pool that resides on the sperm tail axoneme. In other systems, by contrast, the sperm-derived axoneme persists longer after fertilization. This is the case in *Drosophila* for instance, where the axoneme remains present throughout embryogenesis, with constituent axonemal proteins gradually diffusing into the cytoplasm (Pitnick and Karr, 1998). What remains of the axoneme is then enveloped by the developing midgut and defaecated after hatching (Pitnick and Karr, 1998). Unlike the axoneme proper, the centrioles contributed by the sperm are maintained in the zygote of most metazoan species, where they function as microtubule organizing centers. Therefore, it is important that axoneme and centrioles are disconnected from one another to enable such different fates (reviewed in Schatten et al, 2011).

Although not motile, the primary cilium is closely related to flagella in likewise deriving from centrioles and harboring 9 doublets of peripheral axonemal microtubules. The primary cilium can also undergo permanent or transient disassembly, including when G0 cells re-enter the cell cycle and undergo another round of replication and division, with centrioles recruiting Peri-Centriolar Material (PCM) to form the centrosomes that ensure bipolar spindle assembly (for review see Patel and Tsiokas, 2021). Thus, disassembly of the primary cilium is expected to be essential for wound healing, including in the liver or the skin, where quiescent and ciliated hepatocytes or keratinocytes presumably need to lose the primary cilium and reclaim centrioles to assemble centrosomes during mitotic divisions to ensure tissue regeneration.

There are several instances in which elimination of centrioles is critical for proper cellular behavior (reviewed in Kalbfuss and Gönczy, 2023). This is the case for instance during oogenesis in metazoan systems, where such elimination is essential to prevent supernumerary centrioles in the zygote, since two centrioles are already contributed by the sperm in most systems. Failure to remove centrioles from the oocyte can result in multipolar spindle assembly and failed embryonic development. In *Drosophila*, centriole removal during oogenesis is modulated by the Polo kinase (Pimenta-Marques et al, 2016). Polo is present at centrioles and is critical to recruit the PCM. When Polo departs from centrioles, the PCM is lost, resulting in subsequent centriole elimination. Experimentally forcing Polo to remain at centrioles leads to PCM retention and to multipolar spindles in the ensuing zygote (Pimenta-Marques et al, 2016). The mechanism of oogenesis centriole elimination appears to be different in *C. elegans*, where Polo-like kinases and the PCM do not modulate the timing of organelle removal (Pierron et al, 2023). The above findings notwithstanding, understanding of the mechanisms governing elimination of axonemes and centrioles remains incomplete.

Moreover, what may have been the original mode of dismantling these elements is unclear. Investigating these processes in an early branching eukaryote might help address these questions.

*Naegleria gruberi* (*Naegleria* hereafter for simplicity) is a free-living freshwater protozoon that belongs to the Discoba taxon, an early branching eukaryotic lineage. *Naegleria* typically exists as an amoeba feeding on bacteria (Fulton, 1993; Schardinger, 1899). In this life form, cells do not possess centrioles or microtubules during interphase. However, a divergent set of tubulin genes is expressed transiently upon mitosis, thus ensuring assembly of a mitotic spindle in the absence of centrioles (Velle et al, 2022; Fulton, 2022). Remarkably, upon starvation, amoebae transform in a stereotyped manner into a flagellate life form within 90 min (reviewed in Fulton, 1993, 1970). During this transformation, one centriole forms de novo initially, and a second centriole assembles shortly thereafter (Dingle and Fulton, 1966; Fritz-Laylin et al, 2016; Fulton and Dingle, 1967, 1971) reviewed in Fulton, 1993). Both centrioles then template a flagellar axoneme each and organize an intracellular microtubule cytoskeleton. Now equipped with two flagella, cells readily swim away. Of particular importance for this work, most flagellated cells then spontaneously revert to the amoeboid state, and in doing so lose their flagella and their intracellular microtubule cytoskeleton (reviewed in Fulton, 1977b). In contrast to the extensively studied amoeboid-to-flagellate transformation, reversion from the flagellated state to the amoeboid state has been scarcely investigated. As a result, the mechanisms by which axonemes are removed and centrioles may be eliminated are not resolved. *Naegleria*'s position on the evolutionary tree of life, in one of the earliest branching lineages from the Last Eukaryotic Common Ancestor (LECA) (Rodríguez-Ezpeleta et al, 2007), makes it a promising system to offer insights into the origin of these fundamental processes. Moreover, *Naegleria gruberi* is a close relative of *Naegleria fowleri*, also known as the "brain eating amoeba" (reviewed in Güémez and García, 2021). No successful treatment is available for people with an *Naegleria fowleri* infection, which is almost invariably lethal. Thus, increasing understanding of *Naegleria* biology might eventually also suggest new diagnostic and therapeutic avenues. In this work, we set out to investigate the flagellate-to-amoeboid reversion in *Naegleria*, with the intent of unraveling the cellular mechanisms through which the flagellar axoneme and the centrioles are eliminated during this process.

## Results

### Cytological description of reversion from flagellates to amoebae in *Naegleria*

We set out to analyze the process through which *Naegleria* reverts from a flagellate to an amoeba (Fig. 1A). In these experiments, food is removed at $t = 0$ min to induce synchronous transformation from amoebae to flagellates. Thereafter, starting at $t = 90$ min, flagellated cells begin reverting towards the amoeboid state. We found that external flagella are lost from the vast majority of cells within 2 h during this reversion process (Fig. 1B), as previously reported (reviewed in Fulton, 1970). Moreover, analyzing the fate of centrioles during reversion using immunofluorescence with antibodies against the centriolar protein Centrin, we found that the fraction of cells harboring centrioles also diminishes within 2 h

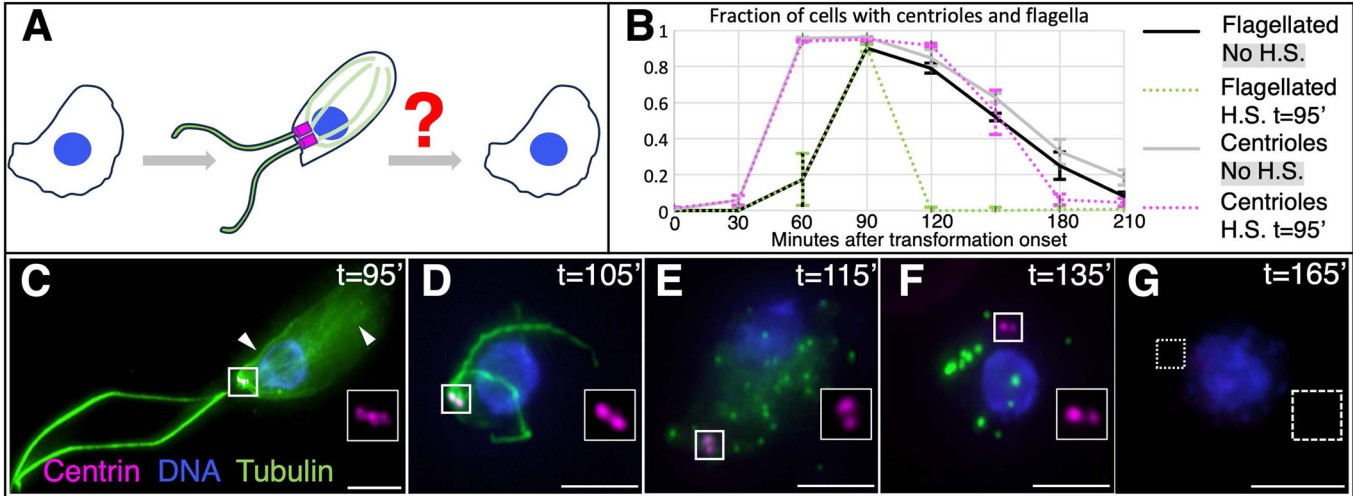

**Figure 1. Reversion process from flagellate to amoeba in *Naegleria gruberi*.**

(A) Schematic of reversion from flagellate to amoeba in *Naegleria*, highlighting the research question. Axonemal and intracellular microtubules: green; centrioles: magenta; nucleus: blue. (B) Fraction of cells with flagella assessed after Lugol's iodine fixation (black), and with centrioles assessed by Centrin immunofluorescence (gray). Corresponding green and magenta dashed lines represent heat-shocked populations (heat shock from $t = 95$ min until $t = 115$ min). Three experimental replicates for each condition. For each replicate, time point and condition: $N > 100$ cells. Error bars represent standard deviations of the mean. Unless stated otherwise, in this and all other figures, $t = 0$ corresponds to the onset of transformation, whereas H.S. refers to heat shock. (C–G) Representative images of cells undergoing reversion after heat shock (from $t = 95$ min until $t = 115$ min). Cells were fixed at indicated times and immunostained for α/β tubulin (green), as well as Centrin (magenta); DNA is visible in blue. Arrowheads in (C) point to intracellular microtubules. Square marks position of inset magnified twofold on the right (Centrin channel); in (G), no centrioles are detected, such that a random area is magnified (indicated by a dashed box). Scale bars are 5 μm. Source data are available online for this figure.

after $t = 90$ min (Figs. 1B and EV1A,B). The reversion process is less synchronous than the transformation process, which presents a challenge for studying the underlying mechanisms. Previous work established that brief exposure to elevated temperature or mechanical pressure can promote reversion (Dingle and Fulton, 1966; Fulton, 1977a; Fulton and Dingle, 1967; Schardinger, 1899). Accordingly, we found that flagellated cells subjected to a 20 min heat shock at $t = 95$ min lose their flagella more synchronously (Fig. 1B). Additionally, centriole elimination is also accelerated, although less so than flagellar loss (Fig. 1B).

To further investigate the elimination of flagella and centrioles, we stained cells at different time points during reversion after heat shock using antibodies against tubulin to mark the axoneme and Centrin to monitor centrioles. At $t = 95$ min, we found that ~90% of cells harbor two flagella stemming from the two centrioles located at the plasma membrane (Fig. 1C). Moreover, an intracellular microtubule cytoskeleton is present below the cell cortex (Fig. 1C, arrowheads). At $t = 105$ min, the flagellar axonemes are internalized in almost all cells, but remain connected to centrioles, whereas intracellular microtubules are absent, although diffuse cytoplasmic signal is detected (Fig. 1D). Thereafter, at $t = 115$ min, most cells no longer contain flagella, but instead exhibit numerous small tubulin-positive spots, as well as diffuse cytoplasmic tubulin signal (Fig. 1E). The tubulin-positive spots then probably coalesce into larger tubulin-positive elements, since at $t = 135$ min focused tubulin signals are less numerous and brighter, whereas the diffuse cytoplasmic tubulin signal has been cleared (Figs. 1F and EV1C,D). Finally, at $t = 165$ min, the majority of cells no longer exhibit signals for tubulin or centrioles (Fig. 1G). In summary, our time-resolved immunofluorescence analysis reveals the sequence of changes in axonemes and centrioles during the reversion process in *Naegleria*.

## Axonemal internalization mechanisms

We focus initially on our analysis of axoneme elimination. Intrigued by the rapid internalization of the flagellar axoneme, we conducted fast time-lapse Differential Interference Contrast (DIC) microscopy of live cells to monitor the internalization process ($N = 30$ cells). Flagellum shedding into the extracellular milieu was never observed in these experiments. Strikingly instead, in 13/30 cells, both flagella fold back laterally onto the cell body and are internalized within milliseconds thereafter (Fig. 2A, white arrows; Movie EV1). This observation is compatible with a hand drawn description reported decades ago (Rafalko, 1947). Moreover, a small bulge on the cell body is usually observed at the site of internalization immediately after contact of the flagellum (Fig. 2A, yellow arrows). In an additional 10/30 cells, the same sequence of events is observed for only one of the two flagella. In 2 of these 10 cases, the other flagellum is retracted inside the cell near-orthogonal to the plasma membrane over the span of several seconds (Fig. EV2A). The fate of the second flagellum could not be ascertained in the other 8 cells of this subset, perhaps because it is out of focus or folds back too rapidly to be observed. In the remaining 7/30 cells analyzed, both flagella were removed, but flagellar internalization could not be conclusively scored; in some cases, a flagellum disappeared between two frames, without any discernible indication of the path it took to presumably be absorbed in the cell.

We reasoned that the extremely rapid disappearance of the flagellum upon folding back laterally onto the cell body, together with the presence of a bulge thereafter, suggests that the flagellum membrane fuses with the plasma membrane of the cell body. To investigate this hypothesis, we treated cells with $MgCl_2$ and $CaCl_2$, which both stimulate membrane fusion in other systems (reviewed in Mondal Roy and Sarkar, 2011). The effect of $MgCl_2$ and $CaCl_2$

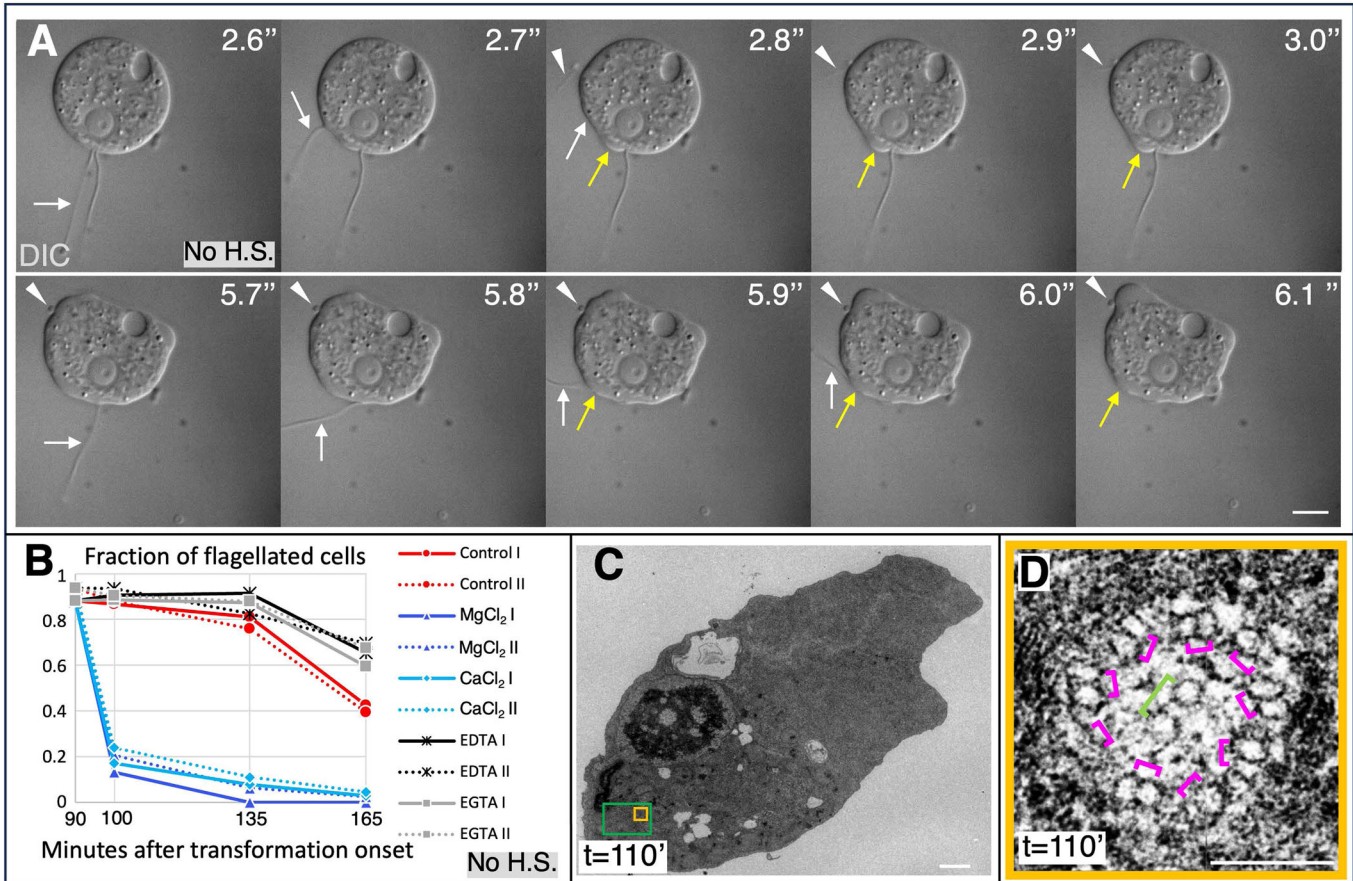

**Figure 2. Axonemal internalization proceeds primarily through membrane fusion.**

(A) Still images of DIC movie of flagellated cell (at -$t = 100$ min). Reversal was induced by the weight of the coverslip, without applying additional pressure. Time is shown in seconds since the beginning of the movie. White arrows point to internalizing flagella, yellow arrows to membrane bulges, arrowheads to a probable remnant of incompletely internalized flagellum. See also Movie EV1. Scale bar is 5 μm. (B) Fraction of cells with flagella assessed after Lugol's iodine fixation. Cells were not heat-shocked. Dashed lines represent experimental repeats for each indicated condition. For each time point and condition: $N > 100$ cells. Note that whereas the effect of Magnesium and Calcium on transformation and reversion was tested previously (Fulton, 1977a), the ions were added at the onset of transformation in those experiments, unlike here. (C) Single section EM from a serial section series, providing an overview of a cell heat shocked a $t = 90$ min and fixed at $t = 110$ min. Orange box indicates region magnified in (D), green box region magnified in Fig. 3E. Scale bar is 1 μm. (D) Single section EM from a serial section series, showing internalized axoneme in top view (corresponding to orange box in (C), without membrane around the 9 microtubule doublets (indicated by magenta brackets, some better seen than others in this particular section). The pair of central microtubules is indicated by a green bracket. Scale bar is 100 nm. Source data are available online for this figure.

on transformation efficiency was previously studied by adding them at the onset of transformation (Fulton, 1977a). By contrast, we added either MgCl$_2$ or CaCl$_2$ when transformation is completed at $t = 90$ min, and found that this accelerates flagellum removal, whereas addition of their chelating agents EDTA or EGTA slows down this process, together compatible with a membrane fusion mechanism (Fig. 2B). To further investigate this possibility, we conducted 50-nm-thick serial section transmission electron microscopy (EM) at $t = 110$ min, when cells transition from having internalized axonemes to harboring small tubulin-positive spots (see Fig. 1D,E; $N = 5$ cells). If axoneme internalization occurs usually via fusion between the membrane of the flagellum and that of the cell body, then internalized axonemes should not be surrounded by a membrane. Serial section EM analysis established that axonemes without a surrounding membrane are indeed present in all cells examined. That this is the case can be observed in side views (Fig. EV2B), as well in rare top views in which

peripheral microtubule doublets and central microtubules can be discerned (Figs. 2D and EV2C). Overall, these experiments indicate that membrane fusion is the principal mechanism underlying internalization of the axoneme at the onset of the reversion process.

## Spastin-mediated axonemal fragmentation

We next analyze how internalized axonemes turn into small tubulin-positive spots (see Fig. 1D,E). Interestingly, immunofluorescence analysis revealed that internalized axonemes exhibit stereotyped discontinuities in the tubulin signal (Fig. 3A). We found that gaps in axonemal tubulin signals are present on average every $0.26 \pm 0.06$ μm (SD), and exhibit a distribution distinct from that expected if fragmentation occurred at random locations (Fig. EV3A). Moreover, fragmentation is observed preferentially at sites with highest axonemal curvature, and therefore highest strain (Fig. 3A, arrowheads; Figs. 3B and EV3B). Furthermore,

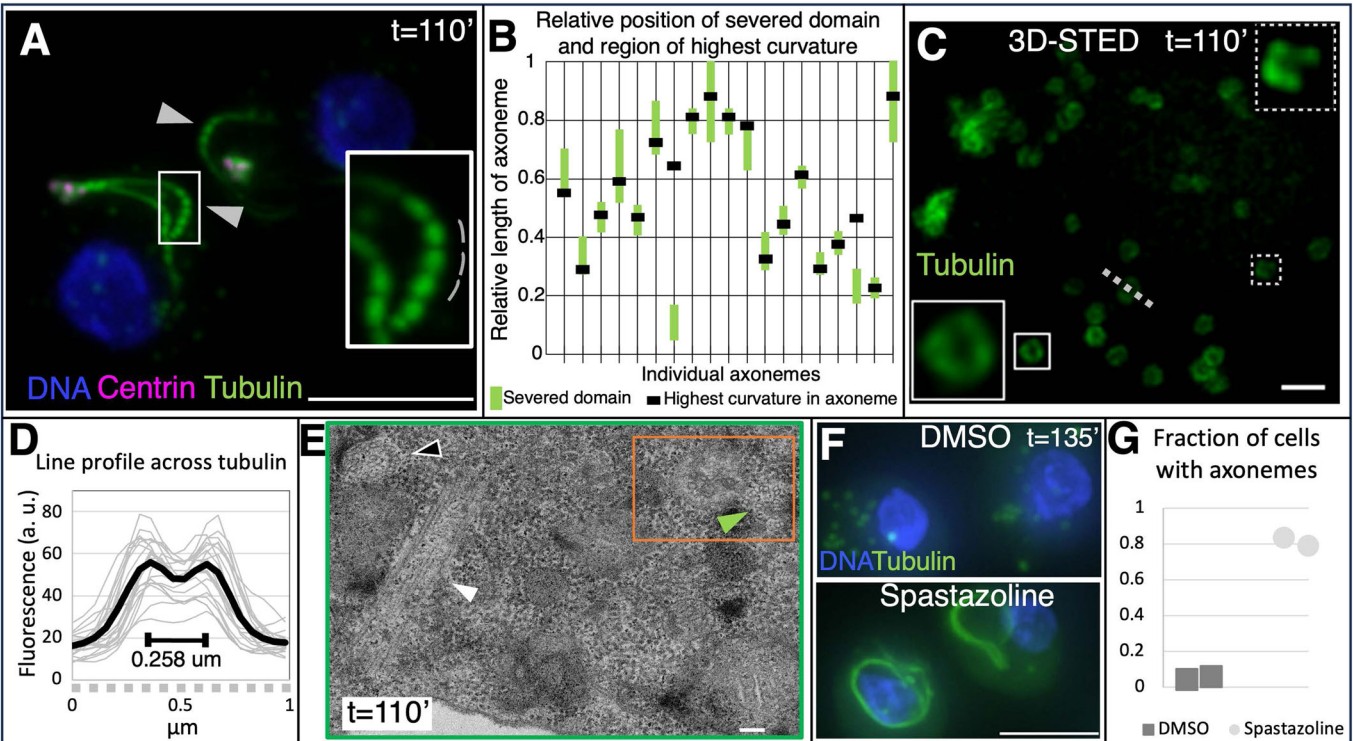

**Figure 3. Internalized axonemes are fragmented primarily by Spastin.**

(A) Cells heat shocked at $t = 95$ min and fixed at $t = 110$ min, immunostained for α/β tubulin (green), as well as Centrin (magenta); DNA is visible in blue. Arrowheads point to the most curved region of the two axonemes in the field of view. White box: most curved region of one axoneme, magnified on the right. The gray lines illustrate how fragment lengths were determined for Fig. EV3A. Scale bar is 5 μm. (B) Spatial relationship between severed domain and region of highest curvature in 19 axonemes analyzed from 2D projected 3D-stack wide-field images. Represented are the relative length of the axoneme (0: position of centriole, 1: other end of axoneme), as well as the relative position of the severed region determined by fitting a segmented line; the region with highest curvature was determined manually. Note that axonemes may still undergo movements once internalized, so that regions of maximal curvature may change over time. Only axonemes that displayed a single severing domain were analyzed. (C) Max-projected 3D STED image of a cell heat shocked at $t = 95$ min and fixed at $t = 110$ min, immunostained for α/β tubulin (green). The white square highlights a circular structure that is magnified twofold on the bottom left, the white dashed square highlights a structure with two parallel lines likewise magnified on the top right. The gray dashed line indicates how the line profile was determined for (D). Scale bar is 1 μm. (D) Line profiles of circular structures in (C) (gray: individual structures, black: average; $N = 20$ structures). Black bar represents the average width between the two peaks. (E) Single section EM from a serial section series (green box in Fig. 2C). White arrowhead: internalized axonemes in side view; black arrowhead with white contour: internalized axoneme in top view, green arrowhead: axoneme fragment. Dark orange box indicates region magnified in Fig. 4A–C. Scale bar is 100 nm. (F) Cells heat shocked at $t = 90$ min for 20 min and incubated with 1% DMSO (top) or 50 μM Spastazolin (bottom); cells were then fixed at $t = 135$ min and immunostained for α/β tubulin (green), as well as Centrin (magenta); DNA is visible in blue. Scale bar is 5 μm. (G) Corresponding quantification of cells harboring an axoneme >5 μm. Two experimental repeats, $N = 142$ and 84 (DMSO) or 94 and 88 (Spastazolin) cells. Source data are available online for this figure.

analysis with STimulated Emission Depletion (STED) super-resolution microscopy revealed that the small tubulin-positive spots are circular structures with a diameter similar to cross-sectional axonemal dimensions, although a bit larger than such dimensions, possibly reflecting slight measurement inaccuracies and/or distortion following fixation (Fig. 3C,D). In addition, we determined the length of axonemal fragments from EM side views (Fig. EV3C), as well as from rare top views in which consecutive serial sections were examined (Figs. 3E and EV3D), overall revealing an average length of $0.35 \pm 0.18$ μm ($N = 6$ fragments).

Which protein could promote axoneme fragmentation? We reasoned that a microtubule severing enzyme could be involved. Together with Katanin and Fidgetin, Spastin is one of three known microtubule severing enzymes containing AAA ATPase domains (Errico et al, 2002; Roll-Mecak and Vale, 2005; for review see Kuo and Howard, 2021), and the only one for which a specific inhibitor has been developed (Cupido et al, 2019; Kuo and Howard, 2021). The

*Naegleria* genome encodes a predicted Spastin homolog that exhibits high conservation within the nucleotide-binding site of the AAA domain, including residues targeted by the human Spastin inhibitor Spastazoline (Fig. EV3E). As shown in Fig. 3F,G, we found that Spastazoline alters the course of axonemal fragmentation, since ~80% of cells lack axonemal fragments at $t = 135$ min upon drug treatment. Fragments are nevertheless observed in ~20% of cells, perhaps reflecting partial inhibition of *Naegleria* Spastin by Spastazoline, or else partial redundancy with other severing mechanisms. Overall, we conclude that internalized axonemes are severed primarily by Spastin into numerous small axonemal fragments.

## Lysosome-mediated removal of axoneme-derived tubulin-containing elements

Following fragmentation by Spastin, the resulting small tubulin-containing spots presumably merge into the brighter tubulin-

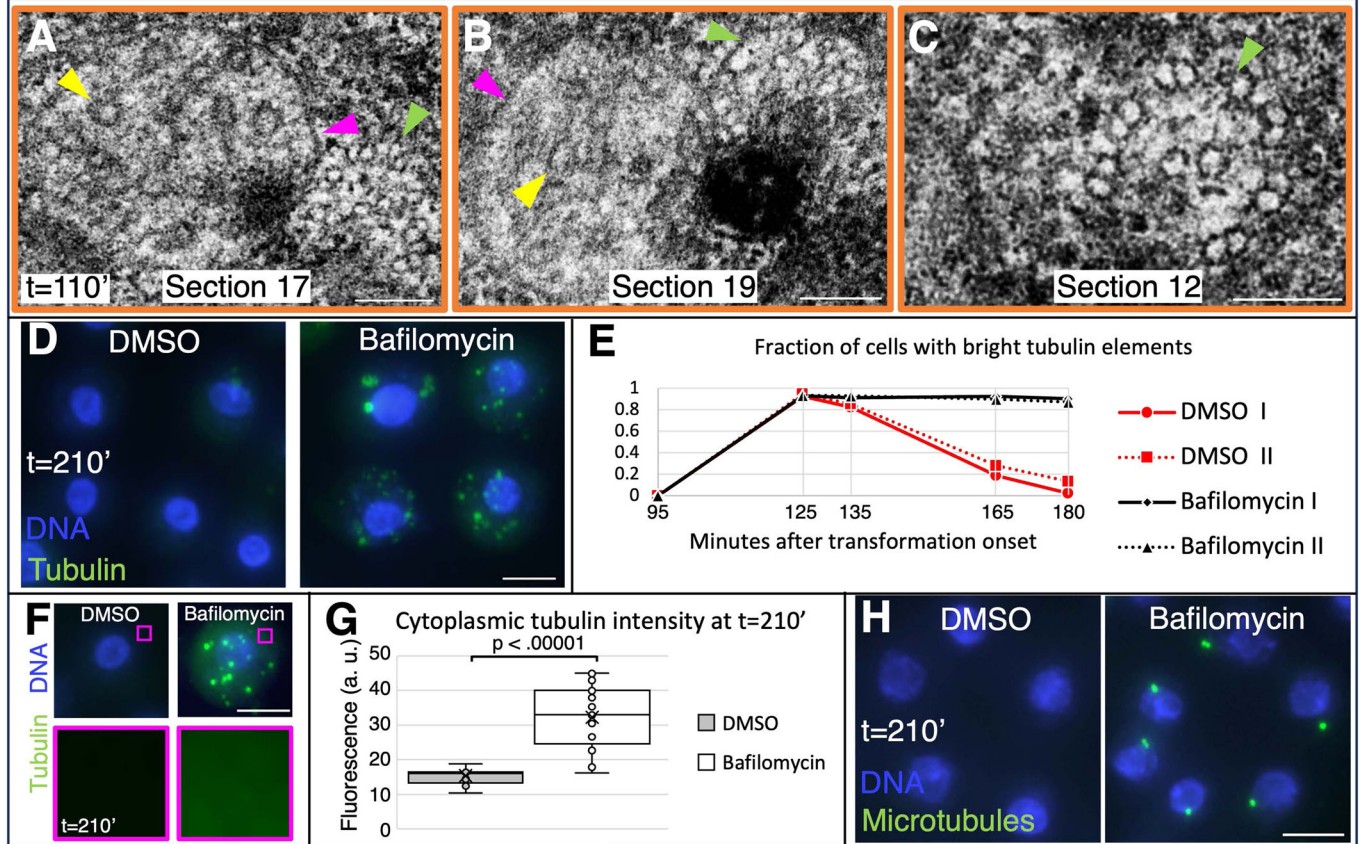

**Figure 4. Tubulin-containing elements are membrane-enclosed and degraded by lysosomes.**

(A–C) Single sections EM from a serial section series (dark orange box in Fig. 3E). Magenta arrowheads: membrane; yellow arrowheads: microtubule; green arrowheads: nearby axoneme fragment (partially out of focus in (A) and with membrane starting to enclose it in (B)). Scale bar is 100 nm. (D) Cells heat shocked for 20 min at $t = 90$ min and incubated with 0.1% DMSO (left) or 10 µM Bafilomycin A1 (right); cells were fixed at $t = 210$ min and immunostained for α/β tubulin (green); DNA is visible in blue. Scale bar is 5 µm. (E) Corresponding quantification of fraction of cells with bright tubulin-containing elements at indicated time points. Dashed lines represent an experimental repeat. $N > 100$ cells for each time point and condition. (F) Same population of cells as in (E). Insets highlight cytoplasmic α/β tubulin signal, which is absent in the control and present in Bafilomycin A1-treated cells. Scale bar is 5 µm. (G) Corresponding quantification of cytoplasmic α/β tubulin signal intensity at $t = 210$ min. $N = 20$ cells for each condition. Boxes represent the interquartile range (IQR), containing the middle 50% of the data. Line represents the median, the cross the mean. Whiskers extend from the first (Q1) and third quartile (Q3) down and up to the smallest and largest value within 1.5 times the IQR below Q1 and above Q3. All data points including outliers are presented. Pairwise comparison by a two-tailed Student's t-tests was conducted, with the resulting $p$ value of 3.712E−10. (H) Cells heat shocked for 20 min at $t = 90$ min and incubated with 0.1% DMSO (left) or 10 µM Bafilomycin A1 (right); cells were fixed at $t = 210$ min and immunostained with the cabazitaxel-derived probe SPY555 marking intact microtubules; DNA is visible in blue. Scale bar is 5 µm. Source data are available online for this figure.

containing elements (see Fig. 1E,F), which we set out to analyze next. Interestingly, serial section EM analysis revealed the presence of membrane-enclosed elements (Fig. 4A, magenta arrowhead), which contain microtubules that are no longer organized in 9 peripheral doublets (Fig. 4A, yellow arrowhead). Moreover, we found that microtubules in such membrane-enclosed elements are wider and more disorganized than those in axonemal fragments devoid of surrounding membranes (Fig. EV4A). Such membrane-enclosed elements could be detected in close proximity of axonemal fragments (Fig. 4B,C, green arrowheads). While most axonemal fragments lack membranes, as mentioned above, in rare cases a membrane could be detected around part of a fragment (Fig. 4B, green arrowhead). Given the presence of membrane-enclosed elements containing wider and more disorganized microtubules, and since the tubulin signal eventually disappears (see Fig. 1G), we hypothesized that these elements correspond to early autophagosomes, which then fuse with lysosomes, thereby resulting in tubulin

removal. To test this hypothesis, we blocked lysosomal acidification using Bafilomycin A1 (Werner et al, 1984). Strikingly, this treatment completely prevents clearance of the bright tubulin-containing elements (Fig. 4D,E). Moreover, this treatment also precludes clearance of the diffuse cytoplasmic tubulin signal (Fig. 4F,G). Addition of Hydroxychloroquine, which prevents the fusion of autophagosomes and lysosomes (Mauthe et al, 2018), has similar effects on the clearance of tubulin-containing elements (Fig. EV4B). Interestingly, in addition, we observed that these elements are not recognized by a cabazitaxel-derived microtubule probe (Fig. 4H), unlike centrioles (see below). This observation suggests that the cabazitaxel binding site is compromised in the slightly wider microtubules present in the putative autophagosomes.

Overall, our findings reveal that axoneme elimination during the reversion from flagellates to amoebae in *Naegleria* follows a characteristic sequence of events. Initially, flagella fold back laterally onto the cell

body and fuse with the plasma membrane within milliseconds, resulting in axoneme internalization. The axoneme is then severed in regions of highest curvature by Spastin, resulting in numerous fragments, which are then enclosed by membranes, presumably generating autophagosomes that fuse with lysosomes, ultimately leading to tubulin degradation through the lysosomal pathway.

## Centrioles are eliminated in lysosomes or in the cytoplasm, and can be shed into the extracellular milieu

We next report our analysis of centriole elimination, which occurs approximately concomitantly with disappearance of the bright tubulin-containing elements (see Fig. 1F,G). High spatial resolution analysis by expansion microscopy coupled to STED (U-Ex-STED) with antibodies against Centrin and tubulin revealed that, compared to intact centrioles at $t = 90$ min (Fig. 5A), centrioles undergoing elimination have lost structural integrity at $t = 150$ min (Fig. 5B). Moreover, in time course experiments conducted without heat shock, we found that centrioles become wider and shorter during elimination (Fig. 5C,D). In addition, centrioles are progressively detached from the plasma membrane and located instead inside the cell at later stages of elimination (Fig. 5E). What mechanisms regulate centriole elimination? To begin investigating this question, we conducted live imaging of this process. Since centriole elimination occurs when cells have already resumed amoeboid motility, thereby preventing imaging of the entire process within a field of view, we blocked amoeboid movements using Latrunculin B to depolymerize the F-actin network (Spector et al, 1983), which did not impair axoneme or centriole elimination (see below). In these experiments, we used a cabazitaxel-derived microtubule probe (added at $t = 0$ min), which marks centrioles and axonemes in *Naegleria* as it does in other systems (Fig. EV4C) (Lukinavičius et al, 2014). Moreover, we provided LysoTracker (added at $t = 120$ min) to mark lysosomes, given their role in removing axonemal tubulin elements in *Naegleria* (see Fig. 4D,E), and various cellular constituents across systems (reviewed in Trivedi et al, 2020). As anticipated, we found that LysoTracker marks lysosomes in *Naegleria*, as evidenced by the signal being contained within membrane-bound entities recognizable by DIC (Fig. EV4D), as well as by the signal being cleared upon Bafilomycin A1 treatment (Fig. EV4E). As shown in Fig. 5F and Movie EV2, we found that in 12/25 cells in which centrioles were labeled with the microtubule probe (~50% of all cells), centrioles are taken up by lysosomes and eliminated within them. In the remaining 13/25 cells, centrioles vanish in the cytoplasm instead (Fig. 5G and Movie EV3). Since cells were immobilized with Latrunculin B in these experiments, it is conceivable that the ratio between the two elimination modes differs when the actin cytoskeleton is intact. For instance, since early steps of autophagy are actin-dependent (for review see: (Kast and Dominguez, 2017)), perhaps a larger fraction of centrioles is eliminated through the lysosomal pathway in unperturbed cells. Alternatively, since Latrunculin B is added in these experiments only at $t = 120$ min, autophagosomes may have already engulfed centrioles by then, in which case the 50:50 ratio would reflect the actual ratio between the two elimination modes.

Strikingly, in addition, whilst conducting shorter term imaging without latrunculin B, we discovered that centrioles can also be shed into the extracellular milieu (Figs. 5H and EV5A). Such externalized

centrioles can be recovered by centrifugation of the culture supernatant onto a coverslip (Fig. EV5B). To estimate the extent to which centriole shedding contributes to organelle removal, we filmed latrunculin B-treated cells, now adding the cabazitaxel-derived microtubule probe only at $t = 130$ min after transformation onset. Such late addition does not allow sufficient time to label centrioles within cells, but the presence of the probe in the culture supernatant enables their detection upon shedding. This analysis established that shedding occurs in ~10% of cells and therefore contributes in a substantial manner to centriole elimination in *Naegleria* (Fig. EV5C; $N = 235$). Moreover, since shedding also occurs in Latrunculin B-treated cells, these experiments establish that this process is actin-independent and thus not driven by exocytosis. What is the fate of centrioles shed into the external milieu? Remarkably, we discovered that externalized centrioles can be taken up by another cell (Figs. 5I and EV5D–F), which is presumably followed by centriole elimination within this new host, given that no centrioles are left in the population 2 h following heat shock (see Fig. 1B).

## The lysosomal pathway and the proteasome mediate centriole elimination

We next set out to test components known to modulate aspects of cellular homeostasis in other systems for their requirement in centriole elimination in *Naegleria*. Given that centrioles vanish in approximately half of the cases within lysosomes, we subjected cells to Bafilomycin A1. Strikingly, this treatment blocked centriole elimination in ~90% of cells (Fig. 6A). Moreover, we found that the number of shed centrioles is significantly higher (Fig. EV5C), presumably reflecting a shift in the balance of organelle elimination modes. Although it remains to be determined why Bafilomycin A1 affects centriole elimination in more cells than the ~50% anticipated from colocalization experiments with LysoTracker (see Fig. 5F), these findings indicate that lysosomal-mediated removal is a major route for centriole elimination in *Naegleria*. In addition, we found that Hydroxychloroquine does not inhibit centriole elimination, in contrast to Bafilomycin A1 (Fig. EV4B), indicative of different mechanisms being at play (see Discussion). Furthermore, given that centrioles are eliminated outside of lysosomes in approximately half of the cases, we also tested whether the proteasome is required by treating cells with the proteasome inhibitor Bortezomib (Fissolo et al, 2008; Lioni et al, 2008). As shown in Fig. 6B, we found that Bortezomib slows down centriole elimination, indicating that proteasome activity normally promotes this process. By contrast, Bortezomib has no impact on the number of shed centrioles (Fig. EV5C). In line with a dual lysosome- and proteasome-mediated mechanism for centriole elimination, we found that centrioles undergoing elimination harbor autophagy-linked K63 polyubiquitination and proteasome-promoting K48 polyubiquitination (Fig. 6C,D) (reviewed in Dósa and Csizmadia, 2022, as well as Grice and Nathan, 2016). Furthermore, we found that ubiquitin moieties are present at centrioles during the elimination process, but not before (compare Fig. 6E–G). Interestingly, in addition, high resolution imaging with U-Ex-STED revealed that Ubiquitin localizes in close proximity to microtubules (Fig. 6H), making them, or proteins closely associated with them, possible targets of ubiquitin-mediated centriole elimination.

Overall, our findings reveal that centriole elimination during reversion from flagellates to amoebae in *Naegleria* entails departure from the cell cortex, followed by elimination primarily in lysosomes

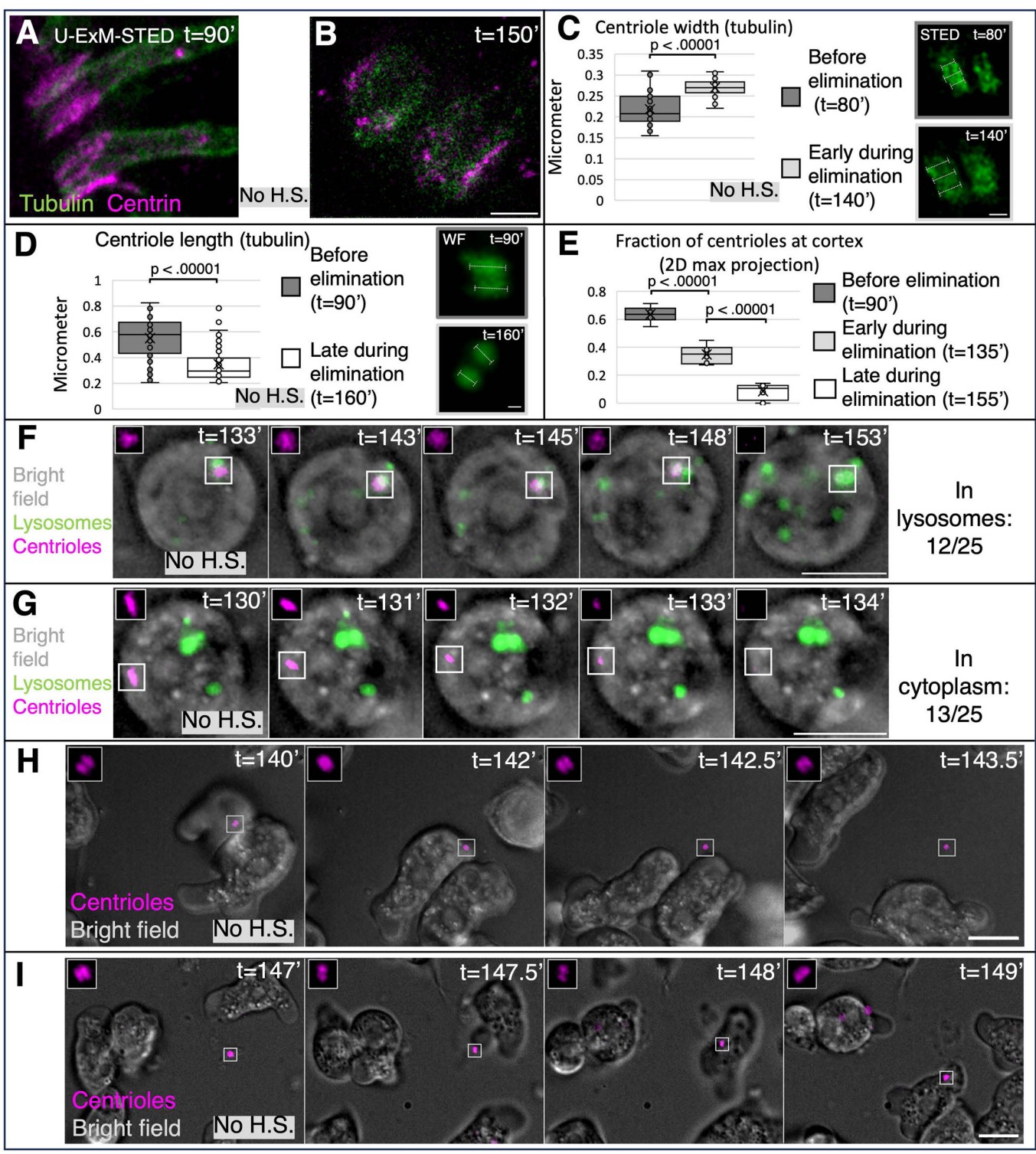

or the cytoplasm. Such elimination is accompanied by ubiquitination in the vicinity of microtubules, with centriole removal mediated by lysosomal and proteasomal activities. In addition, centrioles can be shed into the extracellular milieu and be taken up by other cells before organelle removal.

## Discussion

The mechanisms through which axonemes and centrioles are eliminated upon reversion of *Naegleria* from flagellates to amoebae were not known prior to our work. We uncover that such

**Figure 5. Centrioles are eliminated in lysosomes or the cytoplasm, and can be transferred between cells.**

(A, B) Single plane U-Ex-STED images of centrioles from cells fixed at indicated time points, immunostained for α/β tubulin (green), as well as Centrin (magenta). Scale bar in (B) (for A and B) is 200 nm. (C) Quantification of centriole width (based on α/β tubulin signal) from single-plane STED images, with exemplary images on the right. White lines represent the three measurements for each centriole. $N = 53$ ($t = 80'$) and 40 ($t = 140'$) centrioles. Single-plane STED imaging was used to ensure accurate width measurements. Scale bar is 200 nm. (D) Quantification of centriole length (based on α/β tubulin signal) from projected 3D wide-field images, with exemplary images on the right. White line represents the two measurements for each centriole. $N = 40$ centrioles for each time point. 2D-projections of wide-field Z-stack images were used because we found that single-plane STED microscopy are not ideal for length measurements as centrioles that are not perfectly orthogonal to the lens are imaged only partially. Scale bar is 200 nm. (E) Quantification of localization of centrioles labeled by Centrin antibodies as a function of the cell border ("cortex," visualized by brightfield and unspecific antibody background) in 2D max-projected 3D stacks of widefield microscopy images at indicated time points. Note that manual inspection of 3D images at $t = 90$ min reveals that all centrioles are localized at the cortex, which translates into ~60% in 2D projections. $N = 171$ ($t = 90'$), 190 ($t = 135'$), and 48 ($t = 155'$) cells. (C–E) Boxes represent the interquartile range (IQR), containing the middle 50% of the data. Line represents the median, the cross the mean. Whiskers extend from the first (Q1) and third quartile (Q3) down and up to the smallest and largest value within 1.5 times the IQR below Q1 and above Q3. All data points including outliers are presented. Pairwise comparison by a two-tailed Student's t-tests were conducted, with the resulting $p$ values of 1.5E−12 for C and 7.1E−8 for (D). For (E), one-way ANOVA testing was performed with the resulting $p$ value of 1.568E−11, followed by a Tukey-Kramer post hoc test with the resulting studentized range distributions of T1:T2 $Q = 12.43$ ($p = 0.00000$), T1:T3 $Q = 23.78$ ($p = 0.00000$), and T2:T3 $Q = 11.36$ ($p = 0.00000$). (F, G) Stills of partially 2D-projected 3D stacks of widefield microscopy time-lapse at indicated time points. Amoeboid cells were immobilized with 10 μM latrunculin B at the beginning of the movie, when LysoTracker was added as well; the cabazitaxel-derived microtubule probe SPY650 was present since transformation onset. Reversion to the flagellated state was induced by a 2-min centrifugation at 600 rcf at $t = 95$ min. (F) Cell in which centrioles are degraded in a lysosome. (G) Cell in which centrioles vanish in the cytoplasm. See also Movies EV2 and EV3. Scale bar is 10 μm. (H, I) Stills of a single plane from 3D stacks of widefield microscopy time-lapse at indicated time points. The cabazitaxel-derived microtubule probe SPY650 was present since transformation onset. Reversal to the flagellated state was induced by a 2 min centrifugation at 600 rcf at $t = 95$ min. Panel (G) illustrates centriole shedding, (H) incorporation of that same centriole by another cell (see also Fig. EV5D–F). Note that centriole uptake does not occur in Latrunculin B-treated cells, which cannot move (see Fig. EV5C). Centriole uptake normally occurs in freely moving cells, such that we cannot distinguish whether the lack of uptake in the presence of the drug reflects a direct requirement of actin for uptake, perhaps through endocytosis, or else an indirect requirement of actin for cell movement. Scale bar is 10 μm. Source data are available online for this figure.

elimination occurs in a step-wise fashion, as illustrated schematically in Fig. 6I–N. Flagellum removal usually entails an initial fusion event with the plasma membrane, followed by rapid internalization of the axoneme (Fig. 6J). The axoneme is then severed primarily by Spastin, yielding fragments that are subsequently enclosed by membranes and degraded through the lysosomal pathway (Fig. 6K–M). Centriole elimination entails departure from the plasma membrane, followed by polyubiquitination in the vicinity of microtubules, ultimately resulting in organelle removal within lysosomes or the cytoplasm (Fig. 6L–N). Unexpectedly, we also discovered that centrioles can be shed in the external milieu and then be taken up by other cells (Fig. 6N).

## On flagellar internalization mechanisms

We uncovered that *Naegleria* utilizes two mechanisms for flagellum removal, a main one involving rapid membrane fusion and a minor one involving slower retraction. Having two internalization mechanisms may be optimal to enable flexibility in the face of a changing environment. Moreover, it is conceivable that yet other flagellum removal mechanisms exist when *Naegleria* encounters distinct circumstances or stresses. We established that axonemes are usually internalized by membrane fusion in a low salt liquid culture, and that the pace of axoneme internalization is modulated by bivalent cations. Perhaps retention of at least two potentially ancestral internalization mechanisms reflects adaptive responses to specific environments, for instance varying salt contents or medium viscosity. Diverse modes of flagellum removal in a given organism have been likewise reported in a range of diverse species (Mirvis et al, 2019; Lohret et al, 1998; Rasi et al, 2009; Bloodgood, 1974; Venard et al, 2020). Therefore, together with our findings in the early branching eukaryote *Naegleria*, it is most parsimonious to postulate that multiple mechanisms for flagellum removal coexisted early in eukaryotic evolution.

## On axoneme severing mechanisms

Following internalization in *Naegleria*, axonemes are severed with a regular spacing between cut sites. Pharmacological inhibition experiments attribute this severing activity principally to Spastin. Why do cut sites occur preferentially in regions of highest curvature? Perhaps axonemal microtubules are more readily damaged when under high strain, thereby facilitating Spastin-mediated severing. Another possibility is that Spastin activity is modulated by axonemal curvature, with strain perhaps monitored by Spastin in an analogous manner to how tension is detected at kinetochores by the spindle assembly checkpoint (reviewed in McAinsh and Kops, 2023). For example, a negative regulator of Spastin normally present on axonemes could be removed locally as a consequence of high strain, and thus allow spatially-restricted Spastin activity. Alternatively, Spastin may cut independently of strain, and severing induce the observed curvature. Regardless, the mechanisms by which axonemal fragments are then recognized and surrounded by membranes, eventually resulting in their lysosome-mediated degradation, warrants further investigation. Unlike at centrioles, we did not detect autophagy-promoting K63 poly-ubiquitination at axonemal fragments (Fig. EV6A,B). Perhaps other molecular marks or structural features on these fragments serve as cues for autophagosome formation and eventual lysosomal degradation. Regardless, our findings raise the possibility that axonemal fragments in other systems, for instance those derived from the sperm flagellum in the early mouse embryo (Thompson et al, 1974), are likewise engulfed by membranes before being eliminated.

## On the differential effects of autophagy inhibitors on axonemal and centriole elimination

We found that whereas blocking lysosomal function with Bafilomycin A1 inhibits elimination of both axonemal fragments and centrioles,

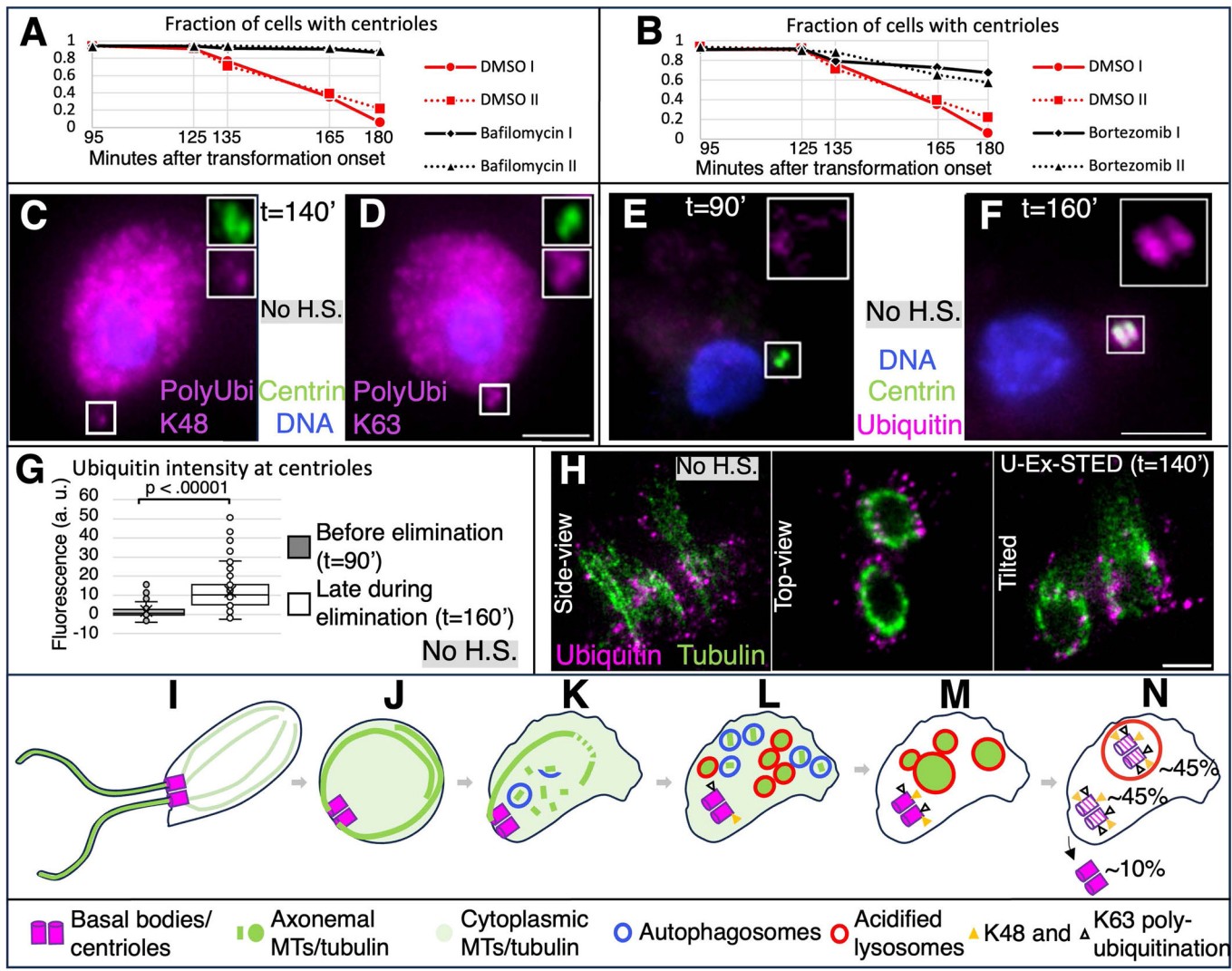

**Figure 6. Centriole elimination depend on lysosome and proteasome function.**

(A, B) Fraction of cells with centrioles as assessed by Centrin immunofluorescence at indicated time points; cells were heat shocked for 20 min at $t = 90$ min. Dashed lines represent experimental repeats. Red: 0.1% DMSO controls; black: 10 μM Bafilomycin A1. (C, D) Cells at $t = 140$ min immunostained with antibodies against Centrin (green), and Poly-ubiquitin K48 (C) or Poly-ubiquitin K63 (D) (both in magenta); DNA is visible in blue. Scale bar in (D) (for C and D) is 5 μm. (E, F) Cells at indicated time points immunostained with antibodies against Centrin (green), as well as ubiquitin (magenta); DNA is visible in blue. Note that poly-ubiquitinated or ubiquitin antibodies generally stain the cytoplasm. However, centrioles can be unambiguously identified as being poly-ubiquitinated when they are lateral to the cell body, as is in (C–F). Scale bar in (F) (for E and F) is 5 μm. (G) Corresponding quantification of ubiquitin signals at centrioles at indicated time points. $N = 67$ ($t = 90'$) or 63 ($t = 160'$) centriole pairs. Boxes represent the interquartile range (IQR), containing the middle 50% of the data. Line represents the median, the cross the mean. Whiskers extend from the first (Q1) and third quartile (Q3) down and up to the smallest and largest value within 1.5 times the IQR below Q1 and above Q3. All data points including outliers are presented. Pairwise comparison by a two-tailed Student's t-tests was conducted, with the resulting $p$ value of 1.741E−11. (H) Single-plane U-Ex-STED images of centrioles in indicated orientations at $t = 140$ min. Scale bar is 200 nm. (I) Flagellated cell with two centrioles and two corresponding flagella. (J) Cell after internalization of axonemes following fusion of flagellar membranes with the plasma membrane of the cell body. (K) Cell undergoing axonemal severing by Spastin at region of highest curvature, and enclosure of resulting fragments by membranes, likely early autophagosomes. (L) Degradation of axonemal fragments in lysosomes. At this stage, centrioles are progressively decorated with poly-ubiquitination chains crosslinked at Lysine 48 and Lysine 63. (M) Fusion of lysosomes and clearance of cytosolic tubulin. (N) Elimination of centrioles either in lysosomes, the cytoplasm, or through externalization, in the indicated proportions. Source data are available online for this figure.

blocking the fusion of autophagosomes with lysosomes with Hydroxychloroquine only prevents axonemal fragments elimination. Bafilomycin A1 inhibits the vacuolar type H+-ATPase, thereby preventing acidification and function of the lysosome (Bowman et al, 1988). By contrast, the molecular target of Hydroxychloroquine is not clear and the compound inhibits autophagosome-lysosome fusion per se (Mauthe et al, 2018). Some forms of autophagy are inhibited by

Bafilomycin A1 but not Hydroxychloroquine, including LC3-associated autophagy (Stempels et al, 2022), a form of macroautophagy (reviewed in Heckmann and Green, 2019). By extension, it is conceivable that *Naegleria* centrioles rely on such a Hydroxychloroquine-insensitive autophagy pathway for elimination. This is compatible with the finding that centrioles, but not axonemal fragments, harbor K63-linked polyubiquitination marks, which is

associated with certain forms of autophagy, including macroautophagy, but not others such as microautophagy (reviewed in Dósa and Csizmadia, 2022).

## Comparing centriole elimination mechanisms during fly and worm oogenesis with *Naegleria*

Comparative analyses of centriole elimination during oogenesis in flies and worms highlighted features specific for each system (Matsuura et al, 2016; Mikeladze-Dvali et al, 2012; Pierron et al, 2023; Pimenta-Marques et al, 2016), which we discuss hereafter in the context of our findings in *Naegleria*. During *Drosophila* oogenesis, experimental retention of the Polo-like kinase 1 (Plk1) Polo at centrioles during meiotic prophase results in PCM maintenance and abrogation of centriole elimination (Pimenta-Marques et al, 2016). Could such a mechanism be at play in *Naegleria*? The genome encodes two predicted kinases with homology to Plk1 (Fritz-Laylin et al, 2010). Whether either protein is recruited to centrioles and serves a shielding role for centrioles remains to be determined. We note that although mitoses occur without centrioles in amoebae (Fulton and Dingle, 1971), in the flagellated life form centrioles seem to organize the intracellular microtubule network, which vanishes early during the reversion process, before centrioles. Therefore, it is intriguing to speculate that here also centrioles might first lose their microtubule-organizing capacity, perhaps by removal of PCM-like components, before the onset of organelle elimination.

We next compare centriole elimination during *C. elegans* oogenesis and *Naegleria* reversion. Centrioles get wider during organelle elimination in both systems, although centrioles do not appear to shorten in the worm (Pierron et al, 2023), in contrast to *Naegleria*, perhaps because worm centrioles are short cube-like structures to start with (Pelletier et al, 2006; Pierron et al, 2023; Woglar et al, 2022; Wolf et al, 1978). In *C. elegans*, the C2-containing microtubule-binding protein SAS-1 regulates the onset of centriole widening during oogenesis (Pierron et al, 2023), and perhaps a related protein exerts an analogous role in *Naegleria*. Organelle widening might be required to promote elimination through the lysosome- or proteasome-mediated pathways, conceivably by providing access to components of the ubiquitination machinery. These two elimination pathways have been investigated in *C. elegans* via downregulation of core components with RNAi, with no apparent impact on organelle demise during oogenesis (Mikeladze-Dvali et al, 2012). However, complete inhibition of these pathways blocks development, so that full chronic depletion cannot be achieved. Acute and complete depletion, for instance through a Degron-mediated system, will be required to probe their implication in the worm with certainty.

## On the regulation and importance of centriole elimination in *Naegleria*

The rapid elimination of the otherwise stable centriole during reversion in *Naegleria* suggests the existence of regulatory mechanisms that enable timely organelle removal. Besides the potential departure of PCM components considered above, given our observation of progressive poly-ubiquitination in the vicinity of microtubules during elimination, controlled expression of E3 ligases could provide a means to initiate organelle removal. We did not identify an E3 ligase upregulated at late times after transformation onset in the existing microarray dataset (Fritz-Laylin and Cande, 2010). Whereas this could reflect insufficient sensitivity of the microarray experiments, alternatively, E3 ligase activity could be controlled post transcriptionally. Why should *Naegleria* ensure that centrioles are removed when cells resume their amoeboid life form? In systems that use centrioles for cell division, their presence is pivotal for bipolar spindle formation and open mitosis, with supernumerary centrioles representing a threat to faithful chromosome segregation. However, as *Naegleria* undergoes vegetative growth with a closed mitotic division, in principle, the presence of two centrioles in the cytoplasm should not interfere with cell division in this case. Nevertheless, the presence of centriole in this life form may result in aberrant cellular organization and function. Preventing centriole elimination may enable to discover such potential roles, and thereby uncover the reason for which centrioles must be removed upon reversion.

In conclusion, our findings establish the detailed sequence of events that entail axoneme and centriole removal in *Naegleria*, and unveil underlying mechanisms that may operate in other systems.

## Methods

**Reagents and tools table**

| Reagent/Resource | Reference or Source | Identifier or Catalog Number |
| --- | --- | --- |
| **Experimental models** | | |
| *Naegleria gruberi* (NEG) | ATTC | ATCC 30223 |
| **Antibodies** | | |
| Mouse anti-α-tubulin conjugated to Alexa 488 | Lima and Cosson, 2019 | AA344 |
| Mouse anti-β-tubulin conjugated to Alexa 488 | Lima and Cosson, 2019 | AA345 |
| Mouse anti-Centrin-2 | Millipore | 04-1624 |
| Mouse anti-Ubiquitin | Enzo | BML-PW0930-0100 |
| Rabbit anti-Ub-K48 | Invitrogen | MA5-35382 |
| Rabbit anti-Ub-K63 | Thermo Fisher | MA5-32573 |
| Goat anti-rabbit conjugated to Alexa Fluor 488 secondary antibodies | Thermo Fisher | A11034 |
| Donkey anti-mouse conjugated to Alexa Fluor 594 secondary antibodies | Abcam | ab150112072 |
| **Chemicals, Enzymes and other reagents** | | |
| Bafilomycin A1 | Thermo Fisher | J61835.MCR |
| Latrunculin B | Merck | L5288 |
| Bortezomib | Focus | 10-2120 |
| Spastazoline | Sigma | SML2659 |
| LysoTracker | Thermo Fisher | LysoTracker™ Green DND-26 |
| Hydroxychloroquine sulfate | Sigma | HO915 |
| SPY650-Tubulin | Spirochrome | SC503 |
| SPY555-Tubulin | Spirochrome | SC203 |

| Reagent/Resource | Reference or Source | Identifier or Catalog Number |
|---|---|---|
| Poly-D-lysine hydrobromide | Merck | P1024 |
| Lugol's iodine solution | Sigma | L6146 |
| Fluoromount-G | Thermo Fisher | F4680 |
| **Software** | | |
| ImageJ | NIH | N/A |

### *Naegleria gruberi* culture and transformation

*Naegleria gruberi* (strain NEG, ATCC 30223) was grown on NM plates (1 liter: 2 g bacto-peptone (Difco), 11.1 mM glucose, 8.97 mM $K_2HPO_4$, 7.34 mM $KH_2PO_4$, 20 g bacto-agar (Difco), autoclaved). A cell population comprising a mixture of amoebae and cysts was maintained on 10 cm diameter NM Petri dishes for 3–7 days at room temperature. For all experiments, cells were washed with water from approximately one-sixth of such a dish, and transferred onto a fresh 10 cm NM Petri dish, along with 500 µl of OP50(BL21) bacteria grown overnight at 37 °C in LB with 50 µg/ml Ampicillin as food source. Cells were distributed evenly with a sterile spreader before incubating them overnight at 33 °C. For initiating the transformation process, 10 ml of 2 mM Tris (not pH-ed) was added to the cells, which were then detached from the Petri dish with a sterile spreader, briefly spun into a pellet with a clinical centrifuge (600 rcf, 45 s), washed 3 times with 10 ml 2 mM Tris each, and incubated at 25.6 °C under agitation (190 rpm) in a 10 ml Erlenmeyer flask. Heat shock was performed at $t = 90$ min or $t = 95$ min after transformation onset, leaving cells at 42 °C under agitation (190 rpm) for the durations indicated in the figure legends.

### Lugol fixation for scoring of flagella

20 µl of Lugol's iodine solution (Sigma) and 20 µl of *Naegleria* culture were mixed on a slide, and covered with a 60 × 24 mm coverslip, followed by immediate analysis using a Nikon Eclipse Ts2 brightfield microscope with a 40× lens.

### Cell collection and fixation for immunofluorescence, U-Ex-STED, and EM

Cells were spun into a pellet with a clinical centrifuge (600 rcf, 1.5 min). For immunofluorescence and U-Ex-STED, cells were resuspended in 1 ml ice-cold (−20 °C) acetone (when using SPY650 in fixed samples) or methanol (for all other immunofluorescence analyses), and fixed for 5 min on ice. This was followed by centrifugation in a table-top centrifuge (750 rcf 1.5 min) at 4 °C and resuspension of the pellet in 1.5 ml PBS. Alternatively, for EM, cells were collected as above but incubated for 1 h in fixation buffer (2% Paraformaldehyde, 2.5% Glutaraldehyde in phosphate buffer 0.1 M pH 7.4). To mount cells on coverslips for immunofluorescence, 8 mm diameter coverslips were washed with ethanol and coated with poly-D-lysine hydrobromide (Merck, P1024); after drying, coverslips were dipped 5 times in water, then 5 times in 100% ethanol, before letting them dry again. Immediately thereafter, each

coverslip was placed in a 1.5 ml Eppendorf tube containing a silicon mould. To prepare such tubes, roughly 750 µl of silicon were added, and the tubes centrifuged at 10,000 rcf for 5 min, resulting in the silicon reshaping so as to provide a surface perpendicular to the centripetal forces of the rotor. Such silicon-containing tubes were left to dry with an open lid for one week before use. Cells in PBS (for immunofluorescence), or in fixation buffer (for EM), were added to such Eppendorf tubes and spun in a table-top centrifuge at room temperature at 600–750 rcf for 1.5 min. Coverslips were stored until use in PBS at 4 °C for immunofluorescence or washed 3 times for 5 min each in cacodylate buffer (0.1 M, pH 7.4) for EM.

### Immunofluorescence and imaging

Coverslips were blocked for 20 min in 3% w/v BSA in PBS-T (PBS (pH 6.8) with 0.1% Tween-20) at room temperature. Primary antibody incubation was performed overnight at 4 °C or for 2 h at room temperature in a moist chamber with primary antibodies diluted in 3% w/v BSA in PBS-T supplemented with 0.05% w/v $NaN_3$. Thereafter, coverslips were washed 3 times for 5 min each in PBS-T prior to incubation with secondary antibodies for 2 h at room temperature in a moist chamber. When antibodies raised in the same species were utilized in a given experiment (for instance antibodies against Centrin and tubulin, both raised in mouse), then one component was first detected as delineated above, before re-incubating the slide with the other directly labeled primary antibody. After three 5 min washes in PBS-T, coverslips were mounted on a slide using 1.5 µl Fluoromount-G (Thermo Fisher), and hardened for 20 min at room temperature in the dark before imaging.

Antibodies used in this study: mouse anti-α-tubulin conjugated to Alexa 488 (1:500, Lima and Cosson, 2019)), mouse anti-β-tubulin conjugated to Alexa 488 (1:500, Lima and Cosson, 2019)), mouse anti-Centrin-2 (1:500, Millipore 04-1624), mouse anti-Ubiquitin (P4D1, 1:500, Enzo, BML-PW0930-0100), rabbit anti-Ub-K48 (ARC0811, 1:500 Invitrogen, MA5-35382), rabbit anti-Ub-K63 (JM09-67, 1:500, Thermo Fisher, MA5-32573), goat anti-rabbit conjugated to Alexa Fluor 488 (1:1000, Thermo Fisher, A11034), as well as donkey anti-mouse conjugated to Alexa Fluor 594 secondary antibodies (1:1000, Abcam, ab150112072).

For labeling of tubulin antibodies with Alexa Fluor 488 or 568 maleimide, 3 equivalents of TCEP (from a 10 mM solution buffered to pH 7) were added at room temperature to the antibodies (typically 0.5 ml from a 1 mg/ml solution in PBS). After 1 h at room temperature, 6 volume equivalents of Alexa Fluor 488 or 568 fluorophore-maleimide in 10 mM DMSO solutions were added to the reduced antibody solution, and the colored mixture incubated at room temperature for 1 h. The labeled antibodies were isolated by gel filtration on a PD-10 column according to the manufacturer's instructions, using PBS during column equilibration and elution. The labeled antibodies were stabilized with 0.01% sodium azide (1:1000 dilution from a 10% aqueous sodium azide stock solution). The degree of labeling (calculated by UV/Vis spectroscopy) was typically between 2 and 3.

Wide-field imaging was performed with a 100×/1.4 Plan-Apochromat objective on a Zeiss Axioplan 2 equipped with a motorized Z-drive (Z steps were 400 nm) and a CoolSnap ES2 camera. 2D-STED images were acquired on a Leica TCS SP8 STED 3X microscope with a 100× 1.4 NA oil-immersion objective, using 488 nm and 589 nm excitation pulsed lasers, as well as 592 nm and

775 nm pulsed lasers for depletion. One-pixel Gaussian blur was applied to all images for analysis and display. For display, brightness and contrast were adjusted in the individual channels, applying the same corrections within a given series.

## Ultrastructure expansion microscopy

Ultrastructure expansion microscopy was performed essentially as reported (Woglar et al, 2022). Methanol-fixed coverslips were washed 3 times in PBS-T for 5 min each, followed by 2 washes in PBS for 5 min each. Coverslips were incubated overnight at room temperature in an Acrylamide/Formaldehyde solution (1% Acrylamide and 1% Formaldehyde in PBS). Thereafter, coverslips were washed 3 times in PBS for 5 min each. For gelation, coverslips were incubated in 50 μl monomer solution (19% (wt/wt) Sodium Acrylate, 10% (wt/wt) Acrylamide, 0.05% (wt/wt) BIS in PBS) supplemented with 0.5% Tetramethylethylenediamine (TEMED) and 0.5% Ammonium Persulfate (APS) on a piece of Parafilm for 1 h at 37 °C in a moist chamber in the dark. Coverslips with gels were incubated in denaturation buffer (200 mM SDS, 200 mM NaCl, and 50 mM Tris in distilled water, pH = 9) until gels detached from the coverslips. Thereafter, gels were incubated in a 1.5 ml Eppendorf tube with denaturation buffer on a heat block at 95 °C. Gels were then transferred to 5 cm Petri dishes, washed with distilled water 5 times for 20 min each, followed by incubation in distilled water overnight at 4 °C. The expansion factor was estimated by measuring the gel size with a ruler, and was typically 4–5 fold.

After expansion, gels were blocked for 1 h in blocking buffer (PBS (pH = 6.8), 3% BSA, 0.1% Tween 20, sodium azide (0.05%)), followed by incubation overnight with primary antibodies diluted in blocking buffer. Gels were washed 3 times in blocking buffer for 10 min each before incubation with secondary antibodies diluted in blocking buffer (supplemented with 0.7 μg/L Hoechst (Bisbenzimide H 33258)) at 37 °C in the dark for 3 h. Gels were washed 3 times in blocking buffer for 10 min each before transfer into a 10 cm Petri dish for re-expansion by washing 6 times for 20 min each in distilled water. For imaging, gels were cut and mounted on a 60 × 24 mm coverslip coated with poly-D-lysine (Sigma, # P1024) diluted in water (2 mg/ml). A second poly-D-lysine-coated coverslip was added on top, and the specimen was sealed with hot VaLaP (1:1:1 mixture of petroleum jelly: lanolin:paraffin wax).

## Live cell imaging

For Fig. 2A, Fig. EV2A, as well as Movie EV1, live cell imaging was performed using DIC microscopy of cells sandwiched between an agarose pad (3% in water) and a coverslip usually 24 × 60 mm in dimensions. Single planes were acquired every 100 ms. Cell movement and stage drift were corrected manually during acquisition. Brightness and contrast were adjusted for display, applying the same corrections within a given series.

For Fig. 5F–I, as well as Fig. EV5A,C,D, inverted live dual bright-field and fluorescence microscopy was performed without a coverslip using a home-made moist chamber mounted on a 40 × 20 mm ethanol-washed coverslip. Imaging was performed with a 63×/1.4 Plan-Apochromat objective on a Nikon Ti2-U equipped with a motorized Z-drive and an Andor Zyla sCMOS camera. Z-steps were 0.5 μm, with time intervals as indicated in the figure legends. All Z-steps in one channel were acquired before imaging in another channel with the same time interval. Brightness and contrast were adjusted for display; for Fig. 5G,F, as well as and Fig. EV5D, a one-pixel Gaussian blur was applied. For Fig. EV5D–F, cell borders were determined by manual inspection of the bright-field channel in multiple optical sections to ensure accurate delimitation.

## Electron microscopy

After washes in cacodylate buffer (see above), coverslips were fixed for 1 h with 2.5% glutaraldehyde and 2.0% paraformaldehyde in 0.1 M phosphate buffer (pH 7.4). Coverslips were then washed three times 5 min each in cacodylate buffer (0.1 M, pH 7.4), and then post-fixed for 40 min in the same buffer with 1.0% osmium tetroxide and 1.5% potassium ferrocyanide, followed by 40 min in 1.0% osmium tetroxide alone. Finally, coverslips were stained for 30 min in 1% uranyl acetate in water, before being dehydrated through increasing concentrations of alcohol, and then finally embedded in epon resin (Embed 812 embedding kit, EMS). Coverslips were placed face down on glass microscope slides with a 2 mm gap between the two glass surfaces filled with resin. This gap was maintained with a 2-mm-thick teflon washer. The samples were then kept in a 65 °C oven for 24 h to allow the resin to harden. Finally, cells were sectioned at 50 nm thickness with a diamond knife (Diatome), and serial sections collected onto pioloform support films on single slot copper grids. These were contrasted with 2% lead citrate and 1% uranyl acetate, and images taken with a transmission electron microscope at 80 kV (Tecnai Spirit, FEI Company with an Eagle CCD camera).

## Pharmacological assays and labels

Compounds were added at the times indicated in the figure legends at the following final concentrations: $MgCl_2$: 10 mM, $CaCl_2$: 10 mM, EDTA: 5 mM, EGTA: 5 mM, Bafilomycin A1: 10 μM (J61835.MCR, Thermo Fisher), Spastazoline (SML2659, Sigma): 50 μM, Latrunculin B (L5288, Merck): 10 μM, Bortezomib (10-2120, Focus): 25 μM, Hydroxychloroquine (HO915, Sigma): 10 μM and 100 μM. LysoTracker (Thermo Fisher, LysoTracker™ Green DND-26) was diluted 1:10,000 for live imaging. SPY650-Tubulin (SC503, Spirochrome) was added at 20 nM for live imaging, SPY555-Tubulin (SC203, Spirochrome) at 20 nM together with antibodies for fixed cell analysis.

## Simulating randomly positioned fragmentation events

To evaluate how distances between fragmentation events would be distributed if severing occurred at random positions along the axoneme, we simulated a random process by choosing positions on a 1D line. We defined the axoneme as a line of length L = 1.5 μm composed of 30 potential cut sites of width dx = 50 nm, reflecting experimental uncertainties in determining the exact position of a cut site. For varying the number of total cuts in the axoneme N, we randomly picked N cut sites from the 30 potential positions and computed the distances between the chosen sites as intercut distances. The distribution of expected intercut distances for N sites was obtained by collecting such distances from 1000 simulations for each N.

## Statistics and quantifications

Two-tailed Student's t-tests and one-way ANOVA test were performed using Prism 10. Tukey-Kramer post hoc tests were performed using www.socscistatistics.com. If not stated otherwise, quantifications are from one representative experiment. Quantifications were conducted in unblinded fashion. Inclusion/exclusion criteria: No data points were omitted.

# Data availability

This study includes no data deposited in external repositories.

The source data of this paper are collected in the following database record: biostudies:S-SCDT-10_1038-S44319-024-00329-w.

# Peer review information

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

## Acknowledgements

We thank Luc Reymond for providing directly labeled tubulin antibodies, as well as Omaya Dudin, Chandler Fulton, Elaine Lai, Marine Olivetta and Anne Villeneuve for constructive comments on the manuscript. We thank the Bioimaging and Optics Core Facility at EPFL for aiding with STED microscopy. This work was supported by grants from the Swiss National Science Foundation (310030_197749) and the European Research Council (AdG 835322) to PG.

## Author contributions

**Alexander Woglar**: Conceptualization; Supervision; Investigation; Methodology; Writing—original draft; Writing—review and editing. **Coralie Busso**: Investigation. **Gabriela Garcia-Rodriguez**: Investigation. **Friso Douma**: Formal analysis. **Marie Croisier**: Investigation. **Graham Knott**: Conceptualization; Supervision. **Pierre Gönczy**: Conceptualization; Supervision; Funding acquisition; Writing—original draft; Writing—review and editing.

Source data underlying figure panels in this paper may have individual authorship assigned. Where available, figure panel/source data authorship is listed in the following database record: biostudies:S-SCDT-10_1038-S44319-024-00329-w.

## Disclosure and competing interests statement

Professor Pierre Gönczy is a member of the Advisory Editorial Board of EMBO Reports. This has no bearing on the editorial consideration of this article for publication.

# Expanded View Figures

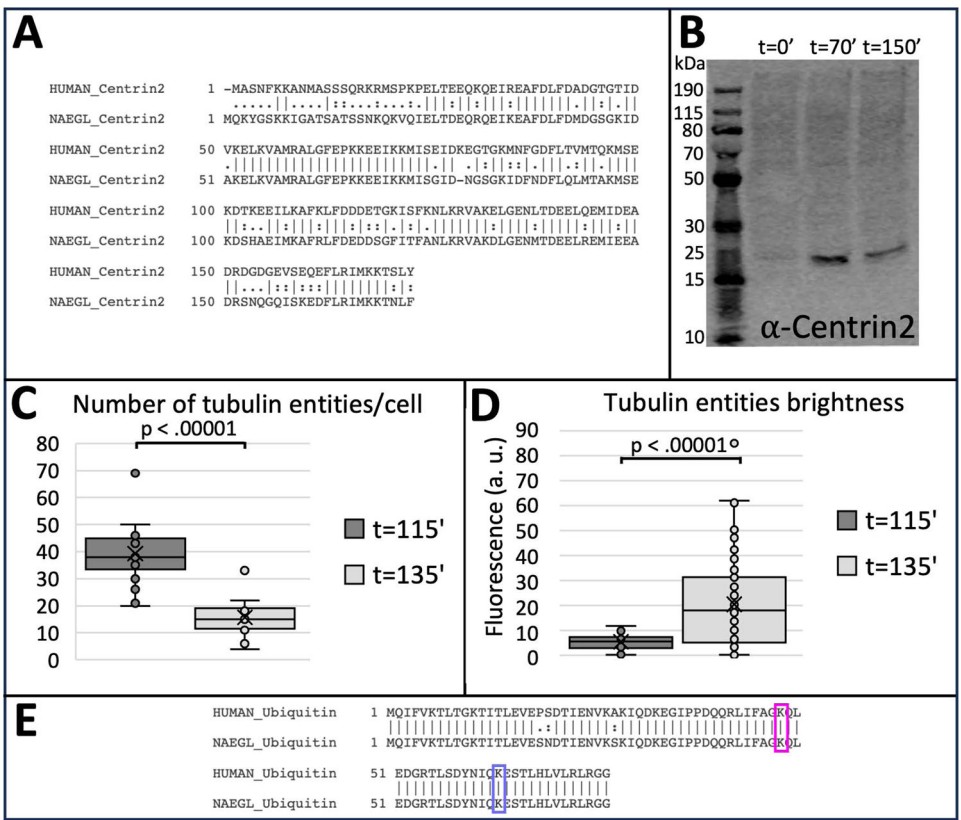

**Figure EV1. Changes in amount and brightness of α/β tubulin spots after axoneme severing.**

(A) ClustalW alignment of human and *Naegleria* Centrin2. (B) Western blot with Centrin antibodies raised against the human protein Centrin-2 on *Naegleria* whole cell lysates harvested at $t = 0$, 70 and 150 min. Note single band at expected size appearing at $t = 70$ min. (C, D) Quantification of number (A, $N = 21$ cell per time point) and brightness (B, $N = 100$ tubulin spots per time point) of α/β tubulin spots per cell at $t = 115$ min and $t = 135$ min after transformation onset, as indicated. Cells were heat shocked from $t = 95$ min until $t = 115$ min. Boxes represent the interquartile range (IQR), containing the middle 50% of the data. Line represents the median, the cross the mean. Whiskers extend from the first (Q1) and third quartile (Q3) down and up to the smallest and largest value within 1.5 times the IQR below Q1 and above Q3. All data points including outliers are presented. Pairwise comparison by a two-tailed Student's t-tests was conducted, with the resulting $p$ values of 3.72E−10 for (C) and 1E−15 for (D). (E) ClustalW alignment of human and *Naegleria* ubiquitin. The K48 (magenta) and K63 (blue) poly-ubiquitinated residues are highlighted.

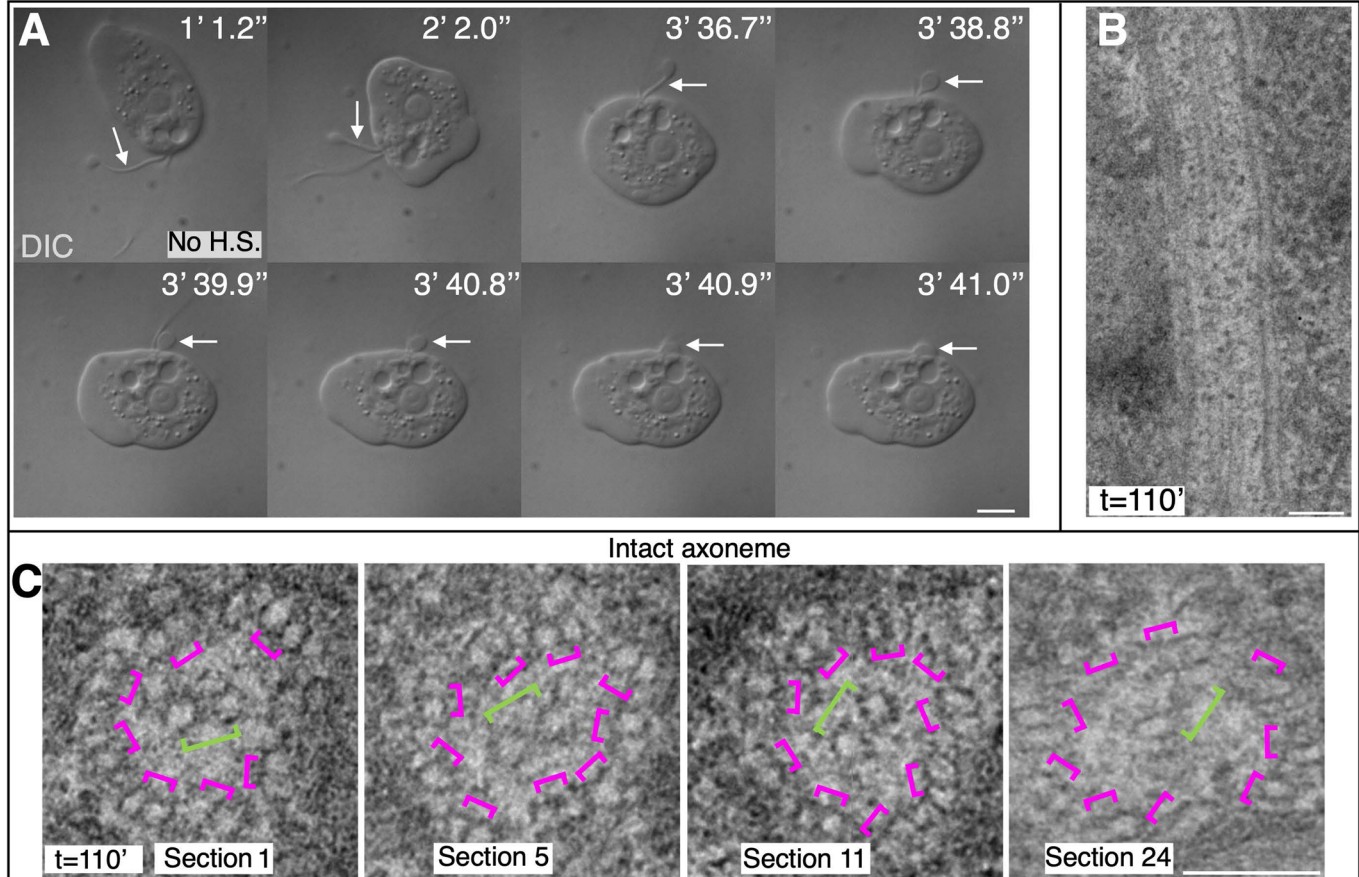

**Figure EV2. Axoneme internalization mechanisms.**

(**A**) Still images of DIC movie of flagellated cell (~$t = 100$ min after transformation onset). Reversal was induced by pressure of the coverslip. Time in min and sec since the beginning of the movie. Arrow: retracting flagellum. Note that a bulge is visible at the end of the retracting flagellum throughout the recording. The second flagellum is present throughout the recording, but out of focus until the first one is completely retracted. In the remainder of the movie, the second flagellum is internalized later by folding back on the cell body and subsequent membrane fusion. Scale bar is 5 μm. (**B**) Single section EM from a serial section series of an internalized axoneme in side view, highlighting the lack of membranes surrounding the microtubule wall at $t = 100$ min after transformation onset. Heat shocked at $t = 90$ min. Scale bar is 100 nm. (**C**) Top views of select 50 nm EM sections from 24 consecutive serial section series from an intact internalized axoneme, from the region shown in Fig. 3E. Microtubule doublets are indicated by magenta brackets, the pair of central microtubules by a green bracket. Note that microtubules are present in all sections (i.e., spanning >1 μm in total). Scale bar is 100 nm.

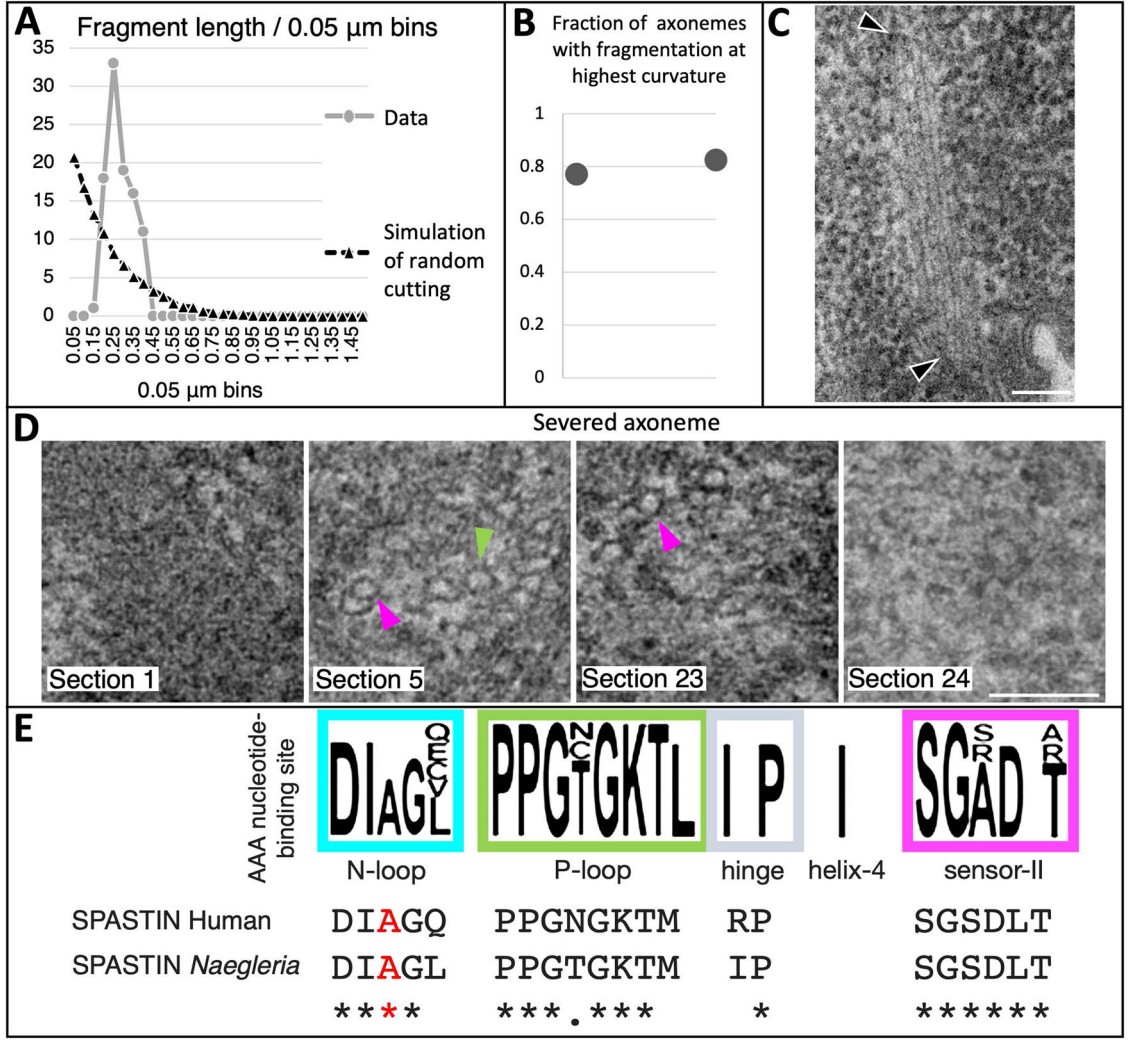

**Figure EV3. Axonemes are severed by Spastin in regions of highest curvature.**

(A) Distribution of fragment length determined in the data (gray discs, binned into 0.05 μm bins; $N = 98$ fragments) or simulated considering random cutting (black triangles). (B) Fraction of axonemes in which the severed region coincides with the region of highest curvature. Two experimental replicates ($N = 74$ and 52 axonemes, respectively). (C) Single section EM from a serial section series of an axonemal fragment in side view. Note that the fragment is ~500-nm long. Black arrowheads indicate the two ends of the fragment. Scale bar is 100 nm. (D) Top views of select 50 nm EM sections from 24 consecutive serial section series from an axonemal fragment, from the region shown in Fig. 3E. Microtubule doublets are indicated by magenta arrowheads, central microtubules by green arrowhead. Note that microtubules are present only between sections 5 and 23 (covering ~900 nm), indicative of this element being larger than the average axonemal fragment. Scale bar is 100 nm. (E) Sequence logo diagram for residues in the nucleotide-binding site of AAA domains that was used to design a specific inhibitor for Spastin (top, from Cupido et al, 2019), as well as alignment of the relevant regions in human and *Naegleria* Spastin (bottom). Spastin Uniprot accession number: D2VS83 (Fritz-Laylin et al, 2010). Note that the N-terminal part of the protein was not correctly annotated and is thus missing in the accession number entry; Uniprot was contacted to correct this. Note also that mutating the "A" in "DIAGO" (highlighted in red) abolishes Spatozoline action on human Spastin (Cupido et al, 2019).

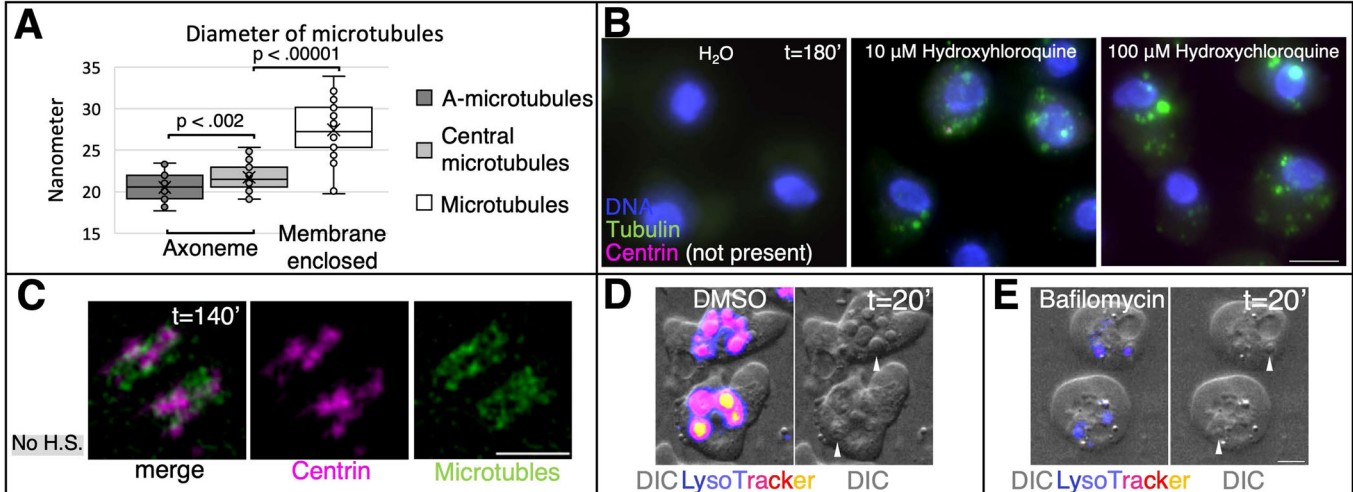

**Figure EV4. Microtubule diameter in membrane enclosed vesicles and axoneme fragments, as well as evidence that LysoTracker labels lysosomes in *Naegleria* and that tubulin positive elements are autophagosome.**

(A) Quantification of microtubule diameters from top views in indicated compartments. Diameters were determined by measuring the perimeter of a fitted circle in individual sections. A-microtubules: $N = 35$ measurements in different sections of 18 microtubules from two axonemes in cross section. Central microtubules: $N = 36$ measurements in different sections of 4 microtubules from two axonemes in cross section. Membrane-enclosed microtubules: $N = 29$ measurements from independent membrane-enclosed elements. Boxes represent the interquartile range (IQR), containing the middle 50% of the data. Line represents the median, the cross the mean. Whiskers extend from the first (Q1) and third quartile (Q3) down and up to the smallest and largest value within 1.5 times the IQR below Q1 and above Q3. All data points including outliers are presented. A one-way ANOVA test was performed with the resulting $p$ value of 1E−15, followed by a Tukey-Kramer post hoc test with the resulting studentized range distributions of T1:T2 Q = 3.09 ($p = 0.07899$), T1:T3 Q = 17.54 ($p = 0.00000$), and T2:T3 Q = 14.45 ($p = 0.00000$). Note that the diameter of the central pair of microtubules in the axoneme is larger than that of the A-microtubule in the peripheral microtubule doublets. (B) Cells heat shocked for 20 min at $t = 95$ min and incubated with 0.1% DMSO (left), 10 μM (middle), or 100 μM Hydroxychloroquine (right); cells were fixed at $t = 180$ min and immunostained for α/β tubulin (green) and Centrin (magenta); DNA is visible in blue. Scale bar is 5 μm. (C) Single plane STED images of centrioles from cells fixed at 105 min, immunostained for Centrin (magenta). Tubulin is visualized with the cabazitaxel-derived microtubule probe Sir-Tubulin (labeled with Rhodamine, pseudo-colored in green). Scale bar is 500 nm. (D, E) Representative single plane DIC and fluorescence images of live cells treated with DMSO or Bafilomycin at $t = 0$ min, imaged at $t = 20$ min after onset of transformation. LysoTracker was added 10 min before imaging. Note that LysoTracker signals can be observed in vesicles. LysoTracker signals are displayed in the ImageJ LUT "Fire", which displays higher gray values in red-yellow and lower gray values in blue-purple color code. White arrowheads indicate examplary vesicles visible by DIC. Scale bar in (E) (for D and E) is 5 μm.

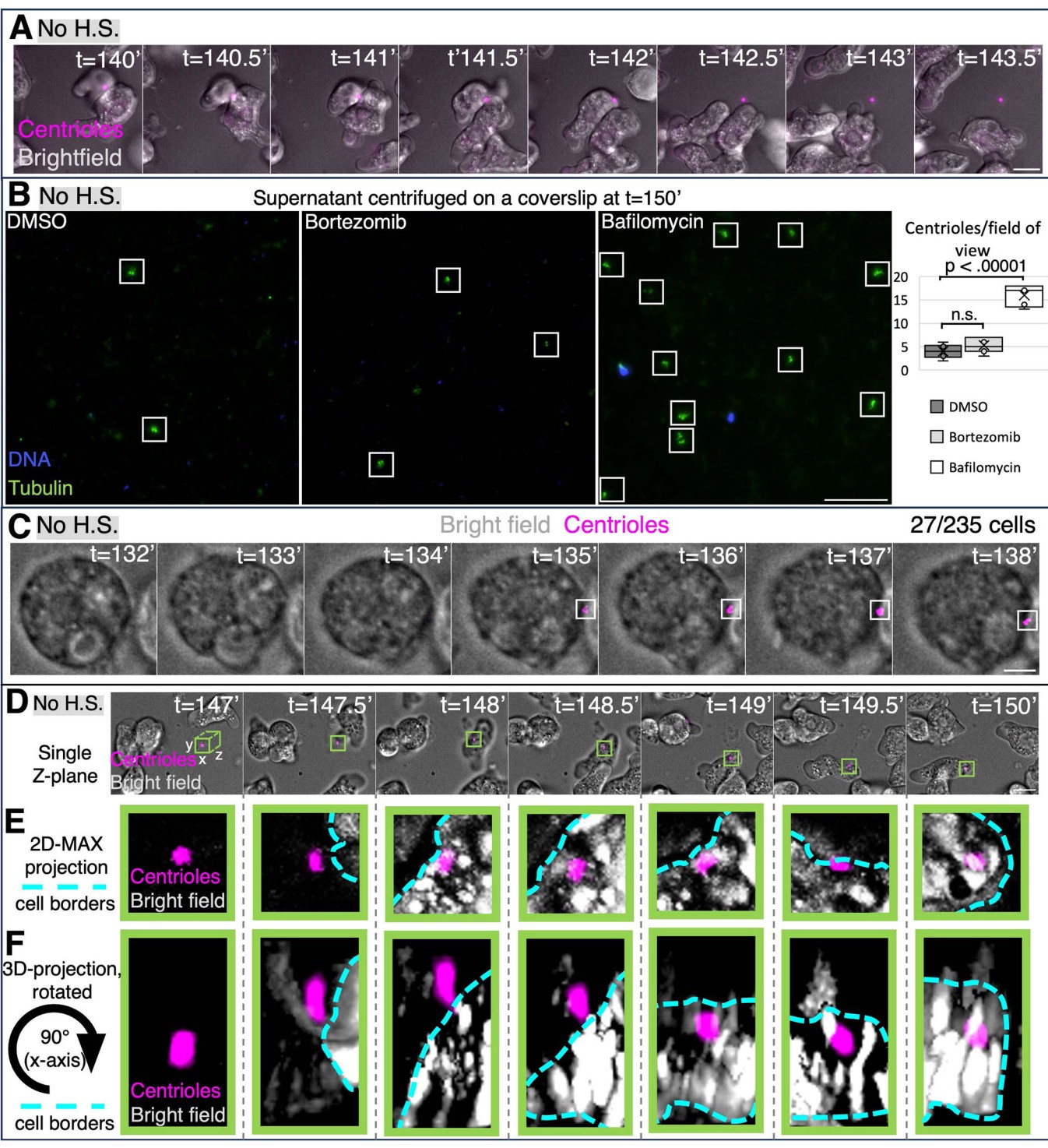

**Figure EV5. Centrioles can be shed in the external environment and taken up by other cells.**

(A) All frames of the movie displayed in Fig. 5H. Time is shown in min since transformation onset. Scale bar is 10 µm. (B) The supernatant of a culture at $t = 150$ min after transformation onset, following a 20 min heat shock at $t = 90$ min, was centrifuged onto a coverslip at 10,000 rcf for 5 min after removal of cells by centrifugation for 2 min at 750 rcf. This was followed by immunostaining for α/β tubulin (green); DNA is visible in blue. $N = 6$ (0.1% DMSO), 7 (25 µM Bortezomib) and 5 (10 µM Bafilomycin A1) fields of view from one representative experiment. (right) Corresponding quantification. Boxes represent the interquartile range (IQR), containing the middle 50% of the data. Line represents the median, the cross the mean. Whiskers extend from the first (Q1) and third quartile (Q3) down and up to the smallest and largest value within 1.5 times the IQR below Q1 and above Q3. All data points including outliers are presented. A one-way ANOVA test was performed with the resulting $p$ value of 128E−8, followed by a Tukey-Kramer post hoc test with the resulting studentized range distributions of T1:T2 Q = 1.79 ($p = 0.43401$), T1:T3 Q = 16.73 ($p = 0.00000$) and T2:T3 Q = 14.94 ($p = 0.00000$). Note that from the ~3-fold increase in centriole shedding estimated from the number of centrioles recovered on coverslips upon Bafilomycin A1 addition, one would expect ~30% of cells lacking centrioles (i.e., 3 times ~10% of cells shedding centrioles). However, this was the case in only ~10% of Bafilomycin A1-treated cells (see Fig. 6A). One possibility to explain this discrepancy is that centrioles shed in control conditions are more fragile than those released upon Bafilomycin A1 treatment, leading to an underestimation when counting those left on the coverslip. Scale bar is 10 µm. (C) Stills of partially 2D-projected 3D stacks of widefield microscopy time-lapse at indicated time points after transformation onset. Amoeboid cells were immobilized with 10 µM Latrunculin B at the beginning of the recording, when LysoTracker and the cabazitaxel-derived microtubule probe SPY650 were added. Note that in this particular case SPY650 was not added at the onset of transformation, which leads to labeling of ~50% of centrioles, but only at the beginning of the movie ($t = 130'$). Thus, centrioles are not detected whilst still in the cell, presumably due to the time it takes SPY650 needs to penetrate the cell and bind centrioles, but become visible upon their shedding (at $t = 135'$). Shedding was observed in 27/235 cells analyzed. Scale bar is 5 µm. (D) Single planes of all frames of the Z-stack movie displayed in Fig. 5I. Green box indicates regions highlighted in (E) and (F). Time in min since transformation onset. Scale bar is 10 µm. (E) Z-max projection of the region boxed in D containing the centriole. (F) 3D projection of the region boxed in (D), rotated by 90° to enable viewing the cell "from the side". Note that until $t = 148.5$ min, centrioles are outside the cell (cell borders, which were determined by visual inspection in 3D, are highlighted by a cyan dashed line in 3D projections at $t = 148$ and 148.5 min; this is visible only in the 3D projection). By contrast, starting at $t = 149$ min, the centrioles are inside the new host.

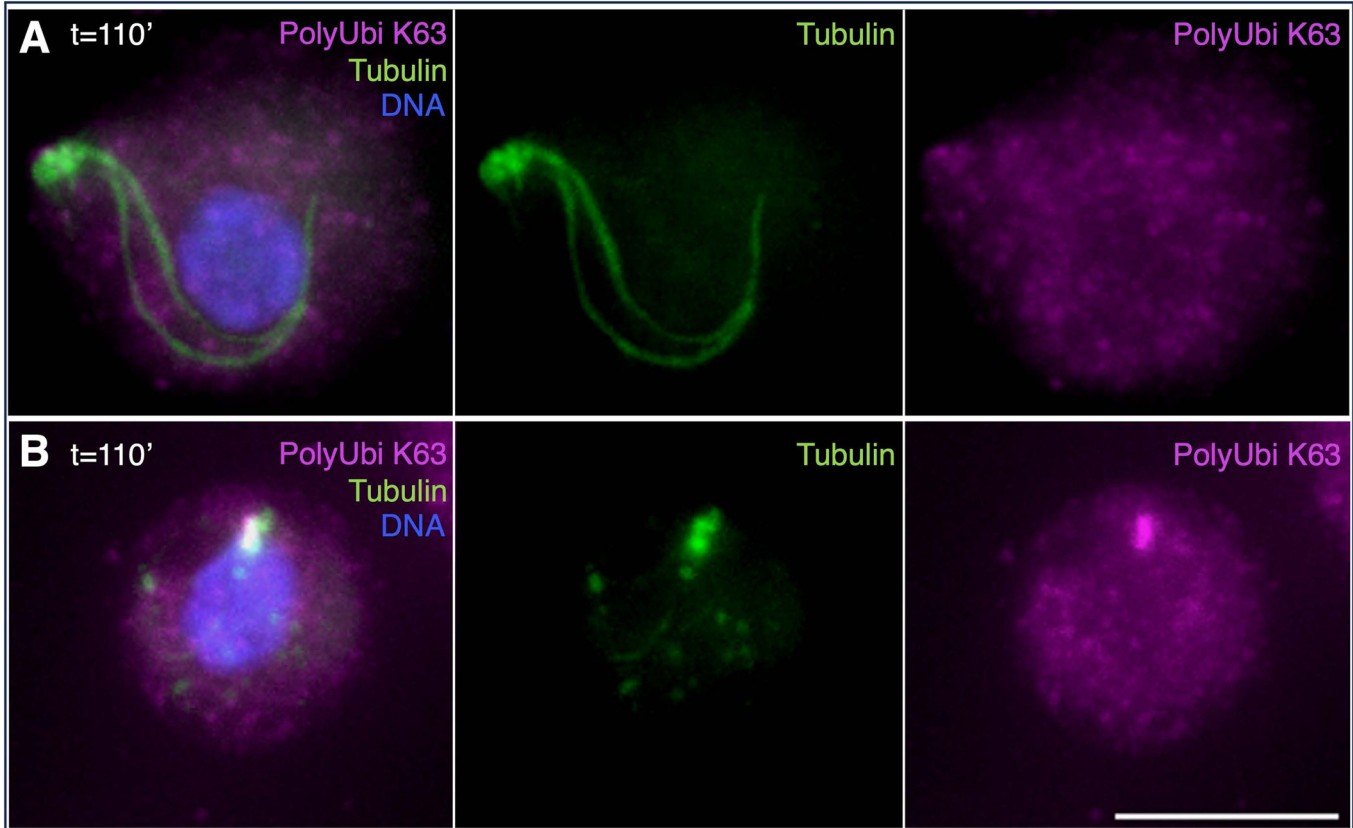

**Figure EV6.   Centrioles, but not flagella or flagellar fragments, harbor K63-linked poly-ubiquitination.**

(A, B) Cells heat shocked at $t = 95$ min and fixed at $t = 110$ min were immunostained with antibodies against α/β tubulin (green), as well as K63-linked poly-ubiquitination (magenta); DNA is visible in blue. The majority of cells at this time point resemble that shown in (A), whereas a minority of them have already undergone axonemal fragmentation (B). Note that K63-linked poly-ubiquitinated antibodies do not mark the axoneme (A) or tubulin positive spots (B), but start to highlight centrioles at the onset of elimination. Scale bar is 5 μm.

