## [Peer Review File · EMBO Reports]

Mechanisms of axoneme and centriole elimination in *Naegleria gruberi*

Pierre Gönczy, Alexander Woglar, Coralie Busso, Gabriela Garcia-Rodriguez, Friso Douma, Marie Croisier, and Graham Knott

Corresponding author(s): Pierre Gönczy (pierre.gonczy@epfl.ch)

Review Timeline:	Transfer Date:	28th Aug 24
	Editorial Decision:	2nd Sep 24
	Revision Received:	5th Sep 24
	Editorial Decision:	4th Oct 24
	Revision Received:	23rd Oct 24
	Accepted:	6th Nov 24

Editor: Deniz Senyilmaz Tiebe

**Transaction Report: This manuscript was transferred to
EMBO reports following peer review at Review Commons.**

Review #1

1. Evidence, reproducibility and clarity:

Evidence, reproducibility and clarity (Required)

In this manuscript the authors dissect the fate of the flagella as a *Naegleria* cell transitions from its flagellate state to an ameboid. The vast majority of work on eukaryotic flagella and cilia focuses on the regulation of their assembly with little on disassembly and loss; therefore, this is an important piece of work as it contributes to our understanding of this understudied area. The authors show that disassembly proceeds through internalisation of the flagellum by fusion of the flagellum membrane with the cell membrane. The axoneme is then broken down through the action of spastin before these fragments are fused with the lysosome. The centriole is lost in parallel through the action of both the proteasome and lysosome. Intriguingly, the authors also see the shedding of the centriole into the media in some cases. Throughout the manuscript the data is well described and the conclusions are supported by the figures. I only have some minor comments that the authors should address.

While I understand the focus is on the loss of flagellum in unicellular organisms but there are a number of more recent reviews about (primary) cilia disassembly that would be worth highlighting here as well e.g. <https://doi.org/10.1042/BCJ20170453> doi.org/10.3390/cells10112977

In addition, flagellum loss in *T. cruzi* occurs through asymmetric division and this alternative method could be added to the introduction as well
DOI:<https://doi.org/10.1016/j.chom.2014.09.003>

The authors in this study rely on commercial antibodies to centrin-2, ubiquitin etc. For the centrin-2 antibody do the authors know if this picks up the expected band by western? The IF is clear but this would be reassuring because as the authors point out *Naegleria* is a divergent eukaryote. On a related note how similar is the ubiquitin sequence to that of the organism the antibody detects? Would you expect a specific signal?

In figure 3C, if the tubulin spots were fragments of the axoneme, I would anticipate these would be randomly oriented in the cell at this point; therefore, not all would show a circular cross section. Do all the spots resolve to circles or are parallel tracks seen?

In figure 3D, it is not clear what the 0.258 μm line is referring to, as it doesn't line up with interpeak distance, which I think it is suppose to. In addition, the authors claim this matches the cross-sectional diameter of an axoneme but that is usually ~ 200 nm so this distance is 25-30% larger.

How often were the membrane enclosed microtubule elements seen by EM? From S4 it looks to be 29 - is this individual sections or 29 independent axoneme containing autophagosomes?

To confirm that cabazitaxel only binds centrioles, do the authors have co-localisation images with the centrin-2 antibody?

In Naegleria, what are the potential mechanisms for uptake? Many are actin dependent and likely affected with latrunculin B treatment. Does endocytosis etc. still occur in these cells?

The addition of bafilomycin resulted in an approximately 3 fold increase in centriole release (S5B) but it had no effect fraction of cells with centrioles. I would expect with a large increase in shedding there would be cells that are centriole negative as they have shed their centrioles but at 150 minutes all cells still have centrioles. Are the centrioles in Naegleria independent or physically connected? Could you have a cell in which one centriole is shed and the other retained?

In Figure 6C/D, what is the general cell staining with the polyubiquitination antibodies? Does this represent the cytoplasm and if so how can you discern the centrioles with this background or is it restricted to just a portion of the cell?

In the discussion should refer to 6I-N.

2. Significance:

Significance (Required)

This is a clear and well-structured manuscript that makes a significant advance to our understanding of flagellum disassembly in Naegleria but also provides an important comparator for work on disassembly mechanisms due to the early-branching nature of this organism, providing a divergent viewpoint from which to interpret flagellum/cilium disassembly in other organisms. Thus, this work will be of interest not only to Naegleria biologists but also those working on cilia and flagella, and the control of organelle loss.

I am a molecular parasitologists with a focus on the flagellum and cytoskeleton of the kinetoplastid parasites.

3. How much time do you estimate the authors will need to complete the suggested revisions:

Estimated time to Complete Revisions (Required)

(Decision Recommendation)

Between 1 and 3 months

4. Review Commons values the work of reviewers and encourages them to get credit for their work. Select 'Yes' below to register your reviewing activity at Web of Science Reviewer Recognition Service (formerly Publons); note that the content of your review will not be visible on Web of Science.

Yes

Review #2

1. Evidence, reproducibility and clarity:

Evidence, reproducibility and clarity (Required)

In the life cycle of *Naegleria*, it undergoes a transition from an amoeboid to a flagellated form, followed by reversion to the amoeboid state. This study elucidates the cellular mechanisms underlying the elimination of flagella and centrioles during this reversion process. The authors employed advanced microscopy techniques, including live and super-resolution imaging, to characterize the cellular processes involved in centriole and flagella removal. The methodology is robust, and the conclusions are substantiated by compelling evidence. This study serves as a cornerstone in further mechanistic exploration of these processes.

****A few minor points:****

1. In the second paragraph of page 3, while a previous report indicated katanin's role in axonemal severing, a recent study demonstrated normal deflagellation after the loss of katanin p60 in a null mutant (Dymek and Smith 2012, JCS, PMID: 22467860).
2. On page 8, it is mentioned that "Flagellum shedding into the extracellular milieu was never observed in these Experiments". However, as shown in movie S1, an isolated flagellum can be observed on the left side of the cell at the time of the first flagellum's disappearance, indicating potential shedding.
3. On page 8, consider removing "of" in "or of CaCl₂".
4. On page 17, the sentence "this could reflect insufficient sensitivity of the microarray experiments or post transcriptional control of E3 ligase activity" seems incomplete. "Sensitivity of post translational control" does not sound right.

2. Significance:

Significance (Required)

The authors have conducted exemplary research to elucidate the cellular processes involved in centriole and flagella removal. This study is foundational, laying the groundwork for further exploration of the underlying molecular mechanisms. Previously, this phenomenon was understood solely at a phenotypic level, but this research has unveiled intricate details of the cell biological processes. However, there appears to be a lack of conceptual advancement, as the molecular mechanisms disclosed are preliminary, without introducing novel concepts or mechanisms.

Researchers interested in the cell biology of microtubule cytoskeleton and/or *Naegleria* would find this work noteworthy.

3. How much time do you estimate the authors will need to complete the suggested revisions:

Estimated time to Complete Revisions (Required)

(Decision Recommendation)

Less than 1 month

4. Review Commons values the work of reviewers and encourages them to get credit for their work. Select 'Yes' below to register your reviewing activity at Web of Science Reviewer Recognition Service (formerly Publons); note that the content of your review will not be visible on Web of Science.

Yes

Review #3

1. Evidence, reproducibility and clarity:

Evidence, reproducibility and clarity (Required)

****Summary:****

Naegleria gruberi cells typically exist as amoebae that lack microtubules, but can differentiate into flagellated cell type—a process which has been relatively well-studied. This flagellate form is transient, however, and not much is known about how cells revert back to their amoeboid state. Here, Woglar et al. investigate this process and provide a model for the elimination of microtubules and centrioles. They find that flagella typically merge with the plasma membrane, are broken into small segments by Spastin, and are degraded by autophagy. While some centrioles are also degraded by autophagy, it seems they can also be degraded in the cytosol by the proteasome, or can be expelled from the cell into the media. Overall, this study is interesting and presents convincing data in a logical, well-thought-out order, and addresses an obvious gap in the literature. I have a few hesitations, however, about the reproducibility of the work, the level of support for some conclusions, and the use of an actin depolymerizing drug.

****Major comments:****

1. It seems most experiments were completed only once or twice, and it is unclear if the quantifications were completed using blinded datasets. This seems okay in some cases where the phenotypes are obvious (Fig. 2B, 3G, 4E, 6A-B). However, because the timing of these experiments is critical to interpreting the data, the experiment in Figure 1B and staining in C-G should be completed a minimum of 3 times. This is essential for some statements including, "we found that flagellated cells subjected to a 20 min heat shock at t=95 min lose their flagella more synchronously (Fig. 1B). Additionally, centriole elimination is also accelerated, although less so than flagellar loss (Fig. 1B)." The first half of this statement is clear from the graph, but the statement about centriole elimination is not well-supported. Perhaps more importantly, in assessing the results of additional experiments, knowing what untreated cells should look like at a given timepoint (e.g. at 120 min for Fig. 4) is important for interpretation.

2. A t-test is not appropriate for comparing more than two samples (e.g. Fig. 5E, S4, S5B).
3. The results shown in Fig. 6C/D are not convincing on their own. Some type of colocalization analysis would strengthen this result. Additionally, showing something that did not localize with the K63 PolyUbi staining would also strengthen this point, and the authors may already have this data on hand-from page 15: "Unlike at centrioles, we did not detect autophagy-promoting K63 poly-ubiquitination at axonemal fragments (not shown)."
4. Please provide evidence for the claim about centriole staining on page 12: "Such late addition does not allow sufficient time to label centrioles within cells, but the presence of the probe in the culture supernatant enables their detection upon shedding." A time lapse to show the labeling process over time would support this point.
5. Is there evidence that LysoTracker is staining the correct compartments in Naegleria? In Fig. S5C, does the cell not have lysosomes, or is that channel missing? The cell in Fig. 5 F seems to accumulate staining over time-could this be a result of phototoxicity?
6. Is there another way to immobilize cells other than treating them with Latrunculin? Because depolymerizing actin will interfere with many other aspects of cell biology, including autophagy (see PMID: 28441569), it seems difficult to interpret some of these results.

****Minor comments:****

1. How were centriole length and width measured at the t=140-160 timepoint for Fig. 5C-D? It is difficult to determine what the measurements would be for the image shown in panel B. Also, why was STED used for C while widefield was used for D?
2. In Fig. 1 B, it is difficult to see the gray circle trace under others
3. Expanding Fig. 1 C-G to show individual channel labeling would be helpful for seeing the centrioles (especially for panel C).
4. In the text describing Fig. 1C, the text notes: "Moreover, an intracellular microtubule cytoskeleton emanates from the two centrioles (Fig. 1C, arrowheads)." While this is certainly reasonable and I agree is very likely the case, is there any evidence this is where the MTs come from? It is not easy to observe from the image shown.
5. In Fig. 1 D, how was it determined that the bundled microtubules are flagella and not collapsed bundles from the cytoplasm? It seems there are more than 2 bundles in the image.
6. In the legend for Fig. 2A, was the "reversal from pushing down on coverslip" from the coverslip settling, or was additional force applied?
7. When discussing Fig. 5 in the main text, it would be helpful to emphasize there was no heat shock.
8. In Fig. S5E-F, how were cell borders determined?
9. There seems to be a minor typo in Fig. 6A, as there are 2 labels for "DMSO I"
10. Chloroquine is widely used to inhibit autophagy in other systems, so the authors may wish to try that here to strengthen their autophagy claims.
11. In the methods, it is stated that SPY-650 tubulin labeling was used for live imaging, but I cannot find this experiment, and it would be amazing to see the experiment shown in Fig. 2A (or any of the timelapse imaging) with this probe!

2. Significance:

Significance (Required)

The strength of this work is that it uncovers basic cell biology of a non-model system, which has a unique lifestyle well-suited to studies about microtubule regulation. The manuscript covers many aspects of the de-flagellation process, and presents a clear, logical model for the disassembly and degradation of axonemes and centrioles. Very little literature exists on this topic, so this is a useful publication that provides important groundwork for future studies on *Naegleria* microtubules. Limitations include that some conclusions are based on cells treated with an inhibitor of actin networks, which may impact the results related to autophagy. This work will be of broad interest to those studying non-model systems, microtubule turnover, or autophagy.

3. How much time do you estimate the authors will need to complete the suggested revisions:

Estimated time to Complete Revisions (Required)

(Decision Recommendation)

Between 1 and 3 months

No

Revision plans

We delineate hereafter in blue fonts the steps that we will take to address the comments made by the three reviewers.

Reviewer #1 (Evidence, reproducibility and clarity (Required)):

In this manuscript the authors dissect the fate of the flagella as a *Naegleria* cell transitions from its flagellate state to an ameboid. The vast majority of work on eukaryotic flagella and cilia focuses on the regulation of their assembly with little on disassembly and loss; therefore, this is an important piece of work as it contributes to our understanding of this understudied area. The authors show that disassembly proceeds through internalisation of the flagellum by fusion of the flagellum membrane with the cell membrane. The axoneme is then broken down through the action of spastin before these fragments are fused with the lysosome. The centriole is lost in parallel through the action of both the proteasome and lysosome. Intriguingly, the authors also see the shedding of the centriole into the media in some cases. Throughout the manuscript the data is well described and the conclusions are supported by the figures. I only have some minor comments that the authors should address.

While I understand the focus is on the loss of flagellum in unicellular organisms but there are a number of more recent reviews about (primary) cilia disassembly that would be worth highlighting here as well

e.g. <https://doi.org/10.1042/BCJ20170453> doi.org/10.3390/cells10112977

> We appreciate that the reviewer understands that our focus is on flagellum loss, and also agree with her/him that a reference regarding primary cilium loss should be mentioned as well.

In addition, flagellum loss in *T. cruzi* occurs through asymmetric division and this alternative method could be added to the introduction as well

DOI:<https://doi.org/10.1016/j.chom.2014.09.003>

> We thank the reviewer for pointing out this interesting paper, which will be referred to in the revised manuscript.

The authors in this study rely on commercial antibodies to centrin-2, ubiquitin etc. For the centrin-2 antibody do the authors know if this picks up the expected band by western? The IF is clear but this would be reassuring because as the authors point out *Naegleria* is a divergent eukaryote. On a related note how similar is the ubiquitin sequence to that of the organism the antibody detects? Would you expect a specific signal?

> Prompted by the suggestion of the reviewer, we will conduct Western blot analysis with Centrin-2 antibodies to test whether they detect a band of the expected size. In addition, we will provide a Supplementary Figure panel with an alignment between human and *Naegleria* Centrin and Ubiquitin.

In figure 3C, if the tubulin spots were fragments of the axoneme, I would anticipate these would be randomly oriented in the cell at this point; therefore, not all would show a circular cross section. Do all the spots resolve to circles or are parallel tracks seen?

> Indeed, as anticipated by the reviewer, fragments are oriented randomly and not all spots resolve to circles. To clarify this point, we will also show a high magnification side view of a fragment in the revised manuscript.

In figure 3D, it is not clear what the 0.258 μm line is referring to, as it doesn't line up with interpeak distance, which I think it is suppose to. In addition, the authors claim this matches the cross-sectional diameter of an axoneme but that is usually ~ 200 nm so this distance is 25-30% larger.

> We are grateful to the reviewer for having spotted that the interpeak distance in Fig. 3D has been inadvertently shifted; this will be rectified in the revised manuscript. Moreover, we will spell out that the determined cross-sectional diameter is indeed ~25% larger than expected, possibly reflecting measurement inaccuracies and/or distortion following fixation.

How often were the membrane enclosed microtubule elements seen by EM? From S4 it looks to be 29 - is this individual sections or 29 independent axoneme containing autophagosomes?

> In the revised manuscript, we will mention explicitly in the legend of Fig. S4 that the 29 instances of membrane-enclosed microtubules indeed correspond to 29 independent measurements of such elements.

To confirm that cabazitaxel only binds centrioles, do the authors have co-localisation images with the centrin-2 antibody?

> As suggested by the reviewer, we will include in the revised manuscript the result of immunofluorescence experiments addressing whether the cabazitaxel probe SPY650 co-localizes with Centrin-2 antibodies.

In Naegleria, what are the potential mechanisms for uptake? Many are actin dependent and likely affected with latrunculin B treatment. Does endocytosis etc. still occur in these cells?

> Assuming that the reviewer refers to the uptake of centrioles, we found indeed that this does not occur in Latrunculin-treated cells. Such cells cannot move, however, and uptake of shed centrioles in the control condition occurs in cells that are freely moving. Therefore, whereas uptake clearly requires intact actin filaments, whether this requirement is direct (i.e. because of failed endocytosis) or indirect (e.g. because of lack of movement) remains to be elucidated. The above points will be discussed explicitly in the revised manuscript.

The addition of bafilomycin resulted in an approximately 3 fold increase in centriole release (S5B) but it had no effect fraction of cells with centrioles. I would expect with a large increase in shedding there would be cells that are centriole negative as they have shed their centrioles but at 150 minutes all cells still have centrioles. Are the centrioles in Naegleria independent or physically connected? Could you have a cell in which one centriole is shed and the other retained?

> This is an interesting point. From the ~3 fold increase in centriole release estimated from the number of centrioles recovered on coverslips upon bafilomycin addition, one would expect ~30% of cells lacking centrioles (i.e. 3 times ~10% of cells that shed centrioles). Instead, we observed this to be the case in only ~10% of cells treated with bafilomycin (Fig. 6A). Whereas the root of this slight discrepancy remains to be elucidated, one possibility is that centrioles shed in control conditions are more fragile than those released upon bafilomycin treatment, leading to an underestimation of their actual numbers when counting those left on the coverslip. That this might be the case will be stated explicitly in the legend of Fig. S5B of the revised manuscript.

In Figure 6C/D, what is the general cell staining with the polyubiquitination antibodies? Does this represent the cytoplasm and if so how can you discern the centrioles with this background or is it restricted to just a portion of the cell?

> As one would expect, poly-ubiquitinated antibodies indeed generally stain the cytoplasm. However, centrioles can be unambiguously identified as being poly-ubiquitinated when they are entirely lateral to the cell body, as in the two figure panels. We will clarify this point in the legend of the figure in the revised manuscript.

In the discussion should refer to 6I-N.

> Thanks for spotting this mislabeling, which will be corrected in the revised manuscript.

Reviewer #1 (Significance (Required)):

This is a clear and well-structured manuscript that makes a significant advance to our understanding of flagellum disassembly in *Naegleria* but also provides an important comparator for work on disassembly mechanisms due to the early-branching nature of this organism, providing a divergent viewpoint from which to interpret flagellum/cilium disassembly in other organisms. Thus, this work will be of interest not only to *Naegleria* biologists but also those working on cilia and flagella, and the control of organelle loss.

I am a molecular parasitologist with a focus on the flagellum and cytoskeleton of the kinetoplastid parasites.

Reviewer #2 (Evidence, reproducibility and clarity (Required)):

In the life cycle of *Naegleria*, it undergoes a transition from an amoeboid to a flagellated form, followed by reversion to the amoeboid state. This study elucidates the cellular mechanisms underlying the elimination of flagella and centrioles during this reversion process.

The authors employed advanced microscopy techniques, including live and super-resolution imaging, to characterize the cellular processes involved in centriole and flagella removal. The methodology is robust, and the conclusions are substantiated by compelling evidence. This study serves as a cornerstone in further mechanistic exploration of these processes.

A few minor points:

1. In the second paragraph of page 3, while a previous report indicated katanin's role in axonemal severing, a recent study demonstrated normal deflagellation after the loss of katanin p60 in a null mutant (Dymek and Smith 2012, JCS, PMID: 22467860).

> Thank you for pointing out this study, which will be referred to in the revised manuscript.

2. On page 8, it is mentioned that "Flagellum shedding into the extracellular milieu was never observed in these Experiments". However, as shown in movie S1, an isolated flagellum can be observed on the left side of the cell at the time of the first flagellum's disappearance, indicating potential shedding.

> We thank the reviewer for her/his careful observation. In fact, the flagellar piece visible on the left side of Movie S1 and Fig. 2A seems to stem from the first flagellum that was probably incompletely internalized following folding back on the plasma membrane. We will point to this flagellar piece in the figure panel and adapt the figure legend accordingly in the revised manuscript.

3. On page 8, consider removing "of" in "or of CaCl₂".

> Thank you for the suggestion; we will do so.

4. On page 17, the sentence "this could reflect insufficient sensitivity of the microarray experiments or post transcriptional control of E3 ligase activity" seems incomplete. "Sensitivity of post translational control" does not sound right.

> We will rephrase the sentence to clarify the intended meaning.

Reviewer #2 (Significance (Required)):

The authors have conducted exemplary research to elucidate the cellular processes involved in centriole and flagella removal. This study is foundational, laying the groundwork for further exploration of the underlying molecular mechanisms. Previously, this phenomenon was understood solely at a phenotypic level, but this research has unveiled intricate details of the cell biological processes. However, there appears to be a lack of

conceptual advancement, as the molecular mechanisms disclosed are preliminary, without introducing novel concepts or mechanisms.

Researchers interested in the cell biology of microtubule cytoskeleton and/or Naegleria would find this work noteworthy.

Reviewer #3 (Evidence, reproducibility and clarity (Required)):

Summary:

Naegleria gruberi cells typically exist as amoebae that lack microtubules, but can differentiate into flagellated cell type—a process which has been relatively well-studied. This flagellate form is transient, however, and not much is known about how cells revert back to their amoeboid state. Here, Woglar et al. investigate this process and provide a model for the elimination of microtubules and centrioles. They find that flagella typically merge with the plasma membrane, are broken into small segments by Spastin, and are degraded by autophagy. While some centrioles are also degraded by autophagy, it seems they can also be degraded in the cytosol by the proteasome, or can be expelled from the cell into the media. Overall, this study is interesting and presents convincing data in a logical, well-thought-out order, and addresses an obvious gap in the literature. I have a few hesitations, however, about the reproducibility of the work, the level of support for some conclusions, and the use of an actin depolymerizing drug.

Major comments:

1. It seems most experiments were completed only once or twice, and it is unclear if the quantifications were completed using blinded datasets. This seems okay in some cases where the phenotypes are obvious (Fig. 2B, 3G, 4E, 6A-B). However, because the timing of these experiments is critical to interpreting the data, the experiment in Figure 1B and staining in C-G should be completed a minimum of 3 times. This is essential for some statements including, "we found that flagellated cells subjected to a 20 min heat shock at t=95 min lose their flagella more synchronously (Fig. 1B). Additionally, centriole elimination is also accelerated, although less so than flagellar loss (Fig. 1B)." The first half of this statement is clear from the graph, but the statement about centriole elimination is not well-supported. Perhaps more importantly, in assessing the results of additional experiments, knowing what untreated cells should look like at a given timepoint (e.g. at 120 min for Fig. 4) is important for interpretation.

> We thank the reviewer for making this important point. As suggested, we will repeat the experiment reported in Fig. 1B and Fig. 1C-G twice more, and adapt Fig. 1B accordingly.

2. A t-test is not appropriate for comparing more than two samples (e.g. Fig. 5E, S4, S5B).

> We realize that this is the case. Accordingly, only pair-wise comparisons were reported for Fig. 5E, S4 and S5B already in the initial submission.

3. The results shown in Fig. 6C/D are not convincing on their own. Some type of colocalization analysis would strengthen this result. Additionally, showing something that did not localize with the K63 PolyUbi staining would also strengthen this point, and the authors may already have this data on hand—from page 15: "Unlike at centrioles, we did not detect autophagy-promoting K63 poly-ubiquitination at axonemal fragments (not shown)."

> We are not certain what additional colocalization analysis the reviewer would like us to perform, since Fig. 6C and 6D already report colocalization between poly-ubiquitin and Centrin. As for the second part of the comment: as suggested, we will document the lack of ubiquitination of axonemal fragments in the revised manuscript.

4. Please provide evidence for the claim about centriole staining on page 12: "Such late addition does not allow sufficient time to label centrioles within cells, but the presence of the probe in the culture supernatant enables their detection upon shedding." A time lapse to show the labeling process over time would support this point.

> In fact, Fig. S5C in the initial submission reports snapshots from a time-lapse sequence of cells imaged following late addition of the cabazitaxel-derived probe, at the beginning of imaging. It is clear that no centrioles were labelled within cells during the entire sequence (in contrast to when we add the probe at the onset of transformation, which leads to labeling of ~50% of centrioles within cells, as mentioned in the manuscript). We will spell out this point in a clearer fashion in the revised manuscript. In addition, should this be needed, we could also provide the movie as a further Supplementary item.

5. Is there evidence that LysoTracker is staining the correct compartments in *Naegleria*? In Fig. S5C, does the cell not have lysosomes, or is that channel missing? The cell in Fig. 5 F seems to accumulate staining over time—could this be a result of phototoxicity?

> LysoTracker is a validated live marker of lysosomes in a broad range of systems, and we therefore expect this to be the case also in *Naegleria*. However, we recognize that this is not absolutely certain, and will therefore conduct further experiments to investigate this question. In particular, we plan to label cells with LysoTracker and address whether the fluorescent signal co-localizes with membrane-enclosed vesicles visible by DIC microscopy. Moreover, we will test whether adding bafilomycin, which inhibits lysosome acidification, prevents the accumulation in these vesicles of LysoTracker, which normally marks lysosomes because of their acidic pH.

6. Is there another way to immobilize cells other than treating them with Latrunculin?

Because depolymerizing actin will interfere with many other aspects of cell biology, including autophagy (see PMID: 28441569), it seems difficult to interpret some of these results.

> We thank the reviewer for her/his suggestion. We have also tried to immobilize *Naegleria* by coating coverslips with collagen, fibronectin or poly-L-lysine, alas without success. Nevertheless, we plan to explore other means of immobilization during the revision process, hoping to find an orthogonal means to immobilize cells. Should this not be feasible, we would spell out explicitly the limitation in the interpretation of our findings stemming from using Latrunculin.

Minor comments:

1. How were centriole length and width measured at the t=140-160 timepoint for Fig. 5C-D? It is difficult to determine what the measurements would be for the image shown in panel B. Also, why was STED used for C while widefield was used for D?

> We apologize for not having been sufficiently clear in describing how length and width were measured for Fig. 5C and 5D. In the revised manuscript, we will explain better how these measurements were conducted and label exemplary images to clarify this point. As for the second question: single-plane STED imaging is optimal to ensure accurate width measurements. As for length measurements, tilted centrioles make it difficult to compare lengths between samples, such that a higher number of specimens needs to be analyzed. Such higher numbers are obtained optimally using wide field microscopy, as many centrioles can be imaged in the same field of view, in contrast to STED microscopy, hence explaining why 2D-projected wide-field images were used instead. These points will be stated explicitly in the revised manuscript.

2. In Fig. 1 B, it is difficult to see the gray circle trace under others

> We realize that the rendition is not ideal, but having tried many other versions, we think that this offers the best visualization. Moreover, there is no ambiguity in understanding the data since the gray circle trace can be clearly identified by magnifying the PDF.

3. Expanding Fig. 1 C-G to show individual channel labeling would be helpful for seeing the centrioles (especially for panel C).

> Thank you for this suggestion, which we will implement in the revised manuscript.

4. In the text describing Fig. 1C, the text notes: "Moreover, an intracellular microtubule cytoskeleton emanates from the two centrioles (Fig. 1C, arrowheads)." While this is certainly reasonable and I agree is very likely the case, is there any evidence this is where the MTs come from? It is not easy to observe from the image shown.

> We agree, and will rephrase the sentence to underscore the fact that the site of microtubule nucleation has not been addressed experimentally.

5. In Fig. 1 D, how was it determined that the bundled microtubules are flagella and not collapsed bundles from the cytoplasm? It seems there are more than 2 bundles in the image.

> We understand how the image in Fig. 1D might have given the impression that there are more than two bundles of microtubules in the cell. This is likely because a maximal Z-projection is shown, and also because the internalized axoneme is curved. Moreover, note that cytoplasmic microtubules have a much smaller diameter than the axoneme. In the revised manuscript, we intend to exchange the figure panel with an image in which out of focus information is less distracting.

6. In the legend for Fig. 2A, was the "reversal from pushing down on coverslip" from the coverslip settling, or was additional force applied?

> No additional force was applied to the coverslip, something that we will convey in the revised manuscript.

7. When discussing Fig. 5 in the main text, it would be helpful to emphasize there was no heat shock.

> Will do so.

8. In Fig. S5E-F, how were cell borders determined?

> Cell borders were determined by manual inspection of the DIC channel, in multiple optical sections, to ensure accurate delimitation. That this is the case will be indicated explicitly in the Materials and Methods section of the revised manuscript.

9. There seems to be a minor typo in Fig. 6A, as there are 2 labels for "DMSO I"

> Thanks for spotting this typo, which will be corrected in the revised manuscript.

10. Chloroquine is widely used to inhibit autophagy in other systems, so the authors may wish to try that here to strengthen their autophagy claims.

> Thank you for this interesting suggestion. We will test the effect of chloroquine, and report the outcome of this experiment in the revised manuscript.

11. In the methods, it is stated that SPY-650 tubulin labeling was used for live imaging, but I cannot find this experiment, and it would be amazing to see the experiment shown in Fig. 2A (or any of the timelapse imaging) with this probe!

> The live imaging experiments (reported in Fig. 5F-I, as well as Movies S2 and S3) indeed used SPY-650 tubulin to mark centrioles, as indicated in the corresponding legends.

Reviewer #3 (Significance (Required)):

The strength of this work is that it uncovers basic cell biology of a non-model system, which has a unique lifestyle well-suited to studies about microtubule regulation. The manuscript covers many aspects of the de-flagellation process, and presents a clear, logical model for

the disassembly and degradation of axonemes and centrioles. Very little literature exists on this topic, so this is a useful publication that provides important groundwork for future studies on *Naegleria* microtubules. Limitations include that some conclusions are based on cells treated with an inhibitor of actin networks, which may impact the results related to autophagy. This work will be of broad interest to those studying non-model systems, microtubule turnover, or autophagy.

Dear Pierre,

Thank you for transferring your manuscript to EMBO Reports, which was previously reviewed at Review Commons.

Referees express interest in the investigation of axoneme and centriole elimination in *Naegleria gruberi*. However, they also raise concerns that need to be addressed to consider publication in EMBO Reports.

Having looked at all documents, we would like to invite you to submit a revised manuscript as in your revision plan. Please revise your manuscript with the understanding that the referee concerns (as in their reports) must be fully addressed and their suggestions taken on board. Please address all referee concerns in a complete point-by-point response. Acceptance of the manuscript will depend on a positive outcome of a second round of review. It is EMBO reports policy to allow a single round of major experimental revision only and acceptance or rejection of the manuscript will therefore depend on the completeness of your responses included in the next, final version of the manuscript.

We realize that it is difficult to revise to a specific deadline. In the interest of protecting the conceptual advance provided by the work, we recommend a revision within 3 months. Please discuss the revision progress ahead of this time with me if you require more time to complete the revisions, or if you have questions or comments regarding the revision (also by video chat).

1. A data availability section providing access to data deposited in public databases is missing (where applicable).
2. Your manuscript contains statistics and error bars based on $n=2$. Please use scatter plots in these cases.

You can submit the revision either as a Scientific Report or as a Research Article. For Scientific Reports, the revised manuscript can contain up to 5 main figures and 5 Expanded View figures, and it should not exceed 27000 characters. If the revision leads to a manuscript with more than 5 main figures it will be published as a Research Article. In this case the Results and Discussion section should be separate. If a Scientific Report is submitted, these sections have to be combined. This will help to shorten the manuscript text by eliminating some redundancy that is inevitable when discussing the same experiments twice. In either case, all materials and methods should be included in the main manuscript file.

4) a .docx formatted letter INCLUDING the reviewers' reports and your detailed point-by-point responses to their comments. As part of the EMBO publication's Transparent Editorial Process, EMBO reports publishes online a Review Process File (RPF) to accompany accepted manuscripts. This File will be published in conjunction with your paper and will include the referee reports, your point-by-point response and all pertinent correspondence relating to the manuscript.

<https://www.embopress.org/page/journal/14693178/authorguide#transparentprocess>

5) a complete author checklist, which you can download from our author guidelines <https://www.embopress.org/page/journal/14693178/authorguide>. Please insert information in the checklist that is also reflected in the manuscript. The completed author checklist will also be part of the RPF.

6) Please note that all corresponding authors are required to supply an ORCID ID for their name upon submission of a revised manuscript (). Please find instructions on how to link your ORCID ID to your account in our manuscript tracking system in our Author guidelines

Additional information on source data and instruction on how to label the files are available:
<https://www.embopress.org/page/journal/14693178/authorguide#sourcedata>

9) Our journal encourages inclusion of *data citations in the reference list* to directly cite datasets that were re-used and obtained from public databases. Data citations in the article text are distinct from normal bibliographical citations and should directly link to the database records from which the data can be accessed. In the main text, data citations are formatted as follows: "Data ref: Smith et al, 2001" or "Data ref: NCBI Sequence Read Archive PRJNA342805, 2017". In the Reference list, data citations must be labeled with "[DATASET]". A data reference must provide the database name, accession number/identifiers and a resolvable link to the landing page from which the data can be accessed at the end of the reference. Further instructions are available at <http://www.embopress.org/page/journal/14693178/authorguide#referencesformat>

12) Please also note our reference format:
<http://www.embopress.org/page/journal/14693178/authorguide#referencesformat>

13) All Materials and Methods need to be described in the main text using our 'Structured Methods' format, which is required for

all research articles. According to this format, the Methods section includes a Reagents and Tools Table (listing key reagents, experimental models, software and relevant equipment and including their sources and relevant identifiers) followed by a Methods and Protocols section describing the methods using a step-by-step protocol format. The aim is to facilitate adoption of the methodologies across labs. More information on how to adhere to this format as well as a downloadable template (.docx) for the Reagents and Tools Table can be found in our author guidelines:
<https://www.embopress.org/page/journal/14693178/authorguide#structuredmethods>.

An example of a Method paper with Structured Methods can be found here:
<https://www.embopress.org/doi/10.15252/msb.20178071>.

I look forward to seeing a revised version of your manuscript when it is ready. Please let me know if you have questions or comments regarding the revision.

Kind regards,

Deniz

Deniz Senyilmaz Tiebe, PhD
Senior Scientific Editor
EMBO Reports

We thank the reviewers for their careful evaluation of our work and their useful suggestions for improving the manuscript. We delineate hereafter in blue fonts how we have addressed all the points that have been raised.

Reviewer #1 (Evidence, reproducibility and clarity (Required)):

In this manuscript the authors dissect the fate of the flagella as a *Naegleria* cell transitions from its flagellate state to an ameboid. The vast majority of work on eukaryotic flagella and cilia focuses on the regulation of their assembly with little on disassembly and loss; therefore, this is an important piece of work as it contributes to our understanding of this understudied area. The authors show that disassembly proceeds through internalisation of the flagellum by fusion of the flagellum membrane with the cell membrane. The axoneme is then broken down through the action of spastin before these fragments are fused with the lysosome. The centriole is lost in parallel through the action of both the proteasome and lysosome. Intriguingly, the authors also see the shedding of the centriole into the media in some cases. Throughout the manuscript the data is well described and the conclusions are supported by the figures. I only have some minor comments that the authors should address.

While I understand the focus is on the loss of flagellum in unicellular organisms but there are a number of more recent reviews about (primary) cilia disassembly that would be worth highlighting here as well

e.g. <https://doi.org/10.1042/BCJ20170453> doi.org/10.3390/cells10112977

> We appreciate that the reviewer understands that our focus is on flagellum loss, and agree with her/him that a reference regarding primary cilium loss should be mentioned as well. This has been included in the revised manuscript (p. 5).

In addition, flagellum loss in *T. cruzi* occurs through asymmetric division and this alternative method could be added to the introduction as well

DOI: <https://doi.org/10.1016/j.chom.2014.09.003>

> We thank the reviewer for pointing out this interesting paper, which we now refer to (p.4).

The authors in this study rely on commercial antibodies to centrin-2, ubiquitin etc. For the centrin-2 antibody do the authors know if this picks up the expected band by western? The IF is clear but this would be reassuring because as the authors point out *Naegleria* is a divergent eukaryote. On a related note how similar is the ubiquitin sequence to that of the organism the antibody detects? Would you expect a specific signal?

> As suggested by the reviewer, we conducted Western blot analysis with the antibodies raised against human Centrin-2, testing *Naegleria* lysates at t=0, t=70 and t=150 min after transformation onset. We found that these antibodies detect a single band of the size expected for *Naegleria* Centrin (~20 kDa), which appears at t=70 min, concomitant with centriole assembly. We provide this new piece of data in Expanded View (EV) 1B of the revised manuscript. In addition, we provide new panels with alignments between human Centrin-2 and *Naegleria* Centrin (EV1A), as well as Ubiquitin (EV1E), highlighting their extremely high degree of conservation.

In figure 3C, if the tubulin spots were fragments of the axoneme, I would anticipate these would be randomly oriented in the cell at this point; therefore, not all would show a circular cross section. Do all the spots resolve to circles or are parallel tracks seen?

> Indeed, as anticipated by the reviewer, fragments are oriented randomly, and not all spots resolve to circles. To clarify this point, we now show also a high magnification side view of a fragment in Fig. 3C of the revised manuscript.

In figure 3D, it is not clear what the 0.258 μm line is referring to, as it doesn't line up with interpeak distance, which I think it is suppose to. In addition, the authors claim this matches the cross-sectional diameter of an axoneme but that is usually ~ 200 nm so this distance is 25-30% larger.

> We are grateful to the reviewer for having spotted that the interpeak distance in Fig. 3D had been inadvertently shifted; this was rectified in the revised manuscript. Moreover, we now spell out that the experimentally determined cross-sectional diameter is $\sim 25\%$ larger than expected, possibly reflecting slightly measurement inaccuracies and/or distortion following fixation (p. 9).

How often were the membrane enclosed microtubule elements seen by EM? From S4 it looks to be 29 - is this individual sections or 29 independent axoneme containing autophagosomes?

> In the revised manuscript, we have mentioned explicitly in the legend of EV4A on page 33 that the 29 instances of membrane-enclosed microtubules indeed correspond to 29 independent measurements of such elements.

To confirm that cabazitaxel only binds centrioles, do the authors have co-localisation images with the centrin-2 antibody?

> Prompted by the suggestion of the reviewer, we collected STED images demonstrating that the cabazitaxel-derived probe indeed marks centrioles (labeled with Centrin-2 antibodies), in addition to internalized axonemes. This piece of data is reported in the new EV4C.

In Naegleria, what are the potential mechanisms for uptake? Many are actin dependent and likely affected with latrunculin B treatment. Does endocytosis etc. still occur in these cells?

> Assuming that the reviewer refers to the uptake of centrioles, we found indeed that this does not occur in Latrunculin B-treated cells. Such cells cannot move, and uptake of shed centrioles normally occurs in freely moving cells. Therefore, we cannot formally distinguish whether the lack of uptake in the presence of the drug reflects a direct requirement of actin for uptake, perhaps through endocytosis, or else an indirect requirement of actin for cell movement. This point is stated explicitly in the legend of Fig. 5 in revised manuscript (p. 30).

The addition of bafilomycin resulted in an approximately 3 fold increase in centriole release (S5B) but it had no effect fraction of cells with centrioles. I would expect with a large increase in shedding there would be cells that are centriole negative as they have shed their centrioles but at 150 minutes all cells still have centrioles. Are the centrioles in Naegleria independent or physically connected? Could you have a cell in which one centriole is shed and the other retained?

> This is an interesting point. From the ~ 3 fold increase in centriole shedding estimated from the number of centrioles recovered on coverslips upon Bafilomycin A1 addition, one would expect $\sim 30\%$ of cells lacking centrioles (i.e. 3 times $\sim 10\%$ of cells shedding centrioles). However, this was the case in only $\sim 10\%$ of Bafilomycin A1-treated cells (see Fig. 6A). One possibility to explain this discrepancy is that centrioles shed in control conditions are more fragile than those released upon Bafilomycin A1 treatment, leading to underestimation when counting those left on the coverslip in control cells. These considerations are mentioned in the revised manuscript (p. 34).

In the discussion should refer to 6I-N.

> Thanks for spotting this mislabeling, which has been corrected (p. 15).

Reviewer #1 (Significance (Required)):

This is a clear and well-structured manuscript that makes a significant advance to our understanding of flagellum disassembly in *Naegleria* but also provides an important comparator for work on disassembly mechanisms due to the early-branching nature of this organism, providing a divergent viewpoint from which to interpret flagellum/cilium disassembly in other organisms. Thus, this work will be of interest not only to *Naegleria* biologists but also those working on cilia and flagella, and the control of organelle loss.

I am a molecular parasitologist with a focus on the flagellum and cytoskeleton of the kinetoplastid parasites.

Reviewer #2 (Evidence, reproducibility and clarity (Required)):

In the life cycle of *Naegleria*, it undergoes a transition from an amoeboid to a flagellated form, followed by reversion to the amoeboid state. This study elucidates the cellular mechanisms underlying the elimination of flagella and centrioles during this reversion process.

The authors employed advanced microscopy techniques, including live and super-resolution imaging, to characterize the cellular processes involved in centriole and flagella removal. The methodology is robust, and the conclusions are substantiated by compelling evidence. This study serves as a cornerstone in further mechanistic exploration of these processes.

A few minor points:

1. In the second paragraph of page 3, while a previous report indicated katanin's role in axonemal severing, a recent study demonstrated normal deflagellation after the loss of katanin p60 in a null mutant (Dymek and Smith 2012, JCS, PMID: 22467860).
> Thank you for pointing out this study, the findings of which have been incorporated in the revised manuscript (p. 3).
2. On page 8, it is mentioned that "Flagellum shedding into the extracellular milieu was never observed in these Experiments". However, as shown in movie S1, an isolated flagellum can be observed on the left side of the cell at the time of the first flagellum's disappearance, indicating potential shedding.
> We thank the reviewer for her/his careful observation. In fact, the flagellar piece visible on the left side of Movie S1 and Fig. 2A seems to stem from the first flagellum that was probably incompletely internalized following folding back onto the plasma membrane. We point to this flagellar piece with an arrowhead in the revised figure panel, and have adapted the figure legend accordingly (p. 26).
3. On page 8, consider removing "of" in "or of CaCl₂".
> Thank you for this suggestion, which has been followed (p. 8).
4. On page 17, the sentence "this could reflect insufficient sensitivity of the microarray experiments or post transcriptional control of E3 ligase activity" seems incomplete. "Sensitivity of post translational control" does not sound right.
> We have rephrased the sentence to clarify the intended meaning (p.18).

Reviewer #2 (Significance (Required)):

The authors have conducted exemplary research to elucidate the cellular processes involved in centriole and flagella removal. This study is foundational, laying the groundwork for further exploration of the underlying molecular mechanisms. Previously, this phenomenon was understood solely at a phenotypic level, but this research has unveiled intricate details of the cell biological processes. However, there appears to be a lack of conceptual advancement, as the molecular mechanisms disclosed are preliminary, without introducing novel concepts or mechanisms.

Researchers interested in the cell biology of microtubule cytoskeleton and/or *Naegleria* would find this work noteworthy.

Reviewer #3 (Evidence, reproducibility and clarity (Required)):

Summary:

Naegleria gruberi cells typically exist as amoebae that lack microtubules, but can differentiate into flagellated cell type-a process which has been relatively well-studied. This flagellate form is transient, however, and not much is known about how cells revert back to their amoeboid state. Here, Woglar et al. investigate this process and provide a model for the elimination of microtubules and centrioles. They find that flagella typically merge with the plasma membrane, are broken into small segments by Spastin, and are degraded by autophagy. While some centrioles are also degraded by autophagy, it seems they can also be degraded in the cytosol by the proteasome, or can be expelled from the cell into the media. Overall, this study is interesting and presents convincing data in a logical, wellthought-out order, and addresses an obvious gap in the literature. I have a few hesitations, however, about the reproducibility of the work, the level of support for some conclusions, and the use of an actin depolymerizing drug.

Major comments:

1. It seems most experiments were completed only once or twice, and it is unclear if the quantifications were completed using blinded datasets. This seems okay in some cases where the phenotypes are obvious (Fig. 2B, 3G, 4E, 6A-B). However, because the timing of these experiments is critical to interpreting the data, the experiment in Figure 1B and staining in C-G should be completed a minimum of 3 times. This is essential for some statements including, "we found that flagellated cells subjected to a 20 min heat shock at t=95 min lose their flagella more synchronously (Fig. 1B). Additionally, centriole elimination is also accelerated, although less so than flagellar loss (Fig. 1B)." The first half of this statement is clear from the graph, but the statement about centriole elimination is not wellsupported. Perhaps more importantly, in assessing the results of additional experiments, knowing what untreated cells should look like at a given timepoint (e.g. at 120 min for Fig. 4) is important for interpretation.

> We thank the reviewer for making this important point. As suggested, we have repeated the experiments reported in Fig. 1B and Fig. 1C-G twice more, and have revised Fig. 1B accordingly.

2. A t-test is not appropriate for comparing more than two samples (e.g. Fig. 5E, S4, S5B).

> We realize that this is the case. Accordingly, only pair-wise comparisons were reported for Fig. 5E, S4 and S5B already in the initial submission.

3. The results shown in Fig. 6C/D are not convincing on their own. Some type of colocalization analysis would strengthen this result. Additionally, showing something that did not localize with the K63 PolyUbi staining would also strengthen this point, and the authors may already have this data on hand-from page 15: "Unlike at centrioles, we did not detect autophagy-promoting K63 poly-ubiquitination at axonemal fragments (not shown)."

> We are not certain what additional colocalization analysis the reviewer would like us to perform, since Fig. 6C and 6D already report colocalization between poly-ubiquitin and Centrin. As for the second part of the comment: as suggested, we now report the lack of K63 poly-ubiquitination of internalized axonemes and axonemal fragments (Fig. S1).

4. Please provide evidence for the claim about centriole staining on page 12: "Such late addition does not allow sufficient time to label centrioles within cells, but the presence of the probe in the culture supernatant enables their detection upon shedding." A time lapse to show the labeling process over time would support this point.

> Fig. S5C in the initial submission shows snapshots from a time-lapse sequence imaged following addition of the cabazitaxel-derived probe late ($t=120$ min), at the beginning of imaging. No centrioles are labelled within cells during the entire sequence in this case, in contrast to what happens when the probe is added at the onset of transformation (with ~50% of centrioles labelled within cells in that case, as mentioned in the manuscript). We have clarified these points in the legend of Fig. EV5C in the revised manuscript (p. 34).

5. Is there evidence that LysoTracker is staining the correct compartments in *Naegleria*? In Fig. S5C, does the cell not have lysosomes, or is that channel missing? The cell in Fig. 5 F seems to accumulate staining over time-could this be a result of phototoxicity?

> Apologies: Figure S5C was labeled incorrectly, as no LysoTracker was present in this experiment; this has been corrected in the revised manuscript. Prompted by the question of the reviewer, we conducted further experiments to verify that LysoTracker is a *bona fide* marker of lysosomes in *Naegleria* as it is in a broad range of systems. Compatible with this view, we found that the LysoTracker fluorescent signal is confined to vesicles clearly detectable by DIC microscopy. Moreover, we found that this signal vanishes when cells are treated with Bafilomycin A1, which inhibits lysosome acidification. These findings are reported in the new figure panels EV4D and EV4E, and establish that LysoTracker is indeed a *bona fide* marker of lysosomes in *Naegleria*.

6. Is there another way to immobilize cells other than treating them with Latrunculin? Because depolymerizing actin will interfere with many other aspects of cell biology, including autophagy (see PMID: 28441569), it seems difficult to interpret some of these results.

> We thank the reviewer for her/his suggestion, which prompted us to attempt immobilizing *Naegleria* differently, including by coating coverslips with collagen, fibronectin or poly-L-lysine, as well as by placing cells into custom-made PDMS micro-chambers, alas all without success. Although we do appreciate the concerns raised by the reviewer, we note that adding Latrunculin B at the onset of filming (at $t=120$ min) enables elimination of ~50% of centrioles via lysosomes. Of course, it is conceivable that the ratio between the two modes of elimination differs when the actin cytoskeleton is intact. We have spelled out explicitly these considerations in the revised manuscript (p. 12).

Minor comments:

1. How were centriole length and width measured at the $t=140-160$ timepoint for Fig. 5C-D? It is difficult to determine what the measurements would be for the image shown in panel B. Also, why was STED used for C while widefield was used for D?

> We apologize for not having been sufficiently clear in describing how centriole length and width were measured. For Fig. 5C, we used single-plane STED imaging to ensure most accurate width measurements. By contrast, for Fig. 5D we used 2D-projections of wide-field Z-stack images because we found that single-plane STED microscopy are not ideal for length measurements as centrioles that are not perfectly orthogonal to the lens are imaged only partially. We explain these points explicitly in the legend of the corresponding figure panels of the revised manuscript (p. 28). In addition, we provide exemplary images of centrioles and illustrate how measurements were conducted in both revised figure panels.

2. In Fig. 1 B, it is difficult to see the gray circle trace under others

> We realize that this rendition is not ideal, but having tried many other possibilities, we think that this offers the best visualization. Moreover, there is no ambiguity in understanding what the data conveys since the gray circle trace can be clearly identified by magnifying the PDF.

3. Expanding Fig. 1 C-G to show individual channel labeling would be helpful for seeing the centrioles (especially for panel C).

> Thank you for this suggestion, which we have implement in the revised manuscript.

4. In the text describing Fig. 1C, the text notes: "Moreover, an intracellular microtubule cytoskeleton emanates from the two centrioles (Fig. 1C, arrowheads)." While this is certainly reasonable and I agree is very likely the case, is there any evidence this is where the MTs come from? It is not easy to observe from the image shown.

> We agree, and have rephrased the sentence to underscore the fact that the site of microtubule nucleation has not been addressed experimentally (p. 7).

5. In Fig. 1 D, how was it determined that the bundled microtubules are flagella and not collapsed bundles from the cytoplasm? It seems there are more than 2 bundles in the image.

> We understand that the original image in Fig. 1D might have given the impression that there are more than two microtubule bundles in the cell, likely because a maximal Z-projection of a cell in which the axoneme curves in an unfavorable orientation was shown. Hence, we exchanged this panel in the revised manuscript, and now show a cell in which the reader can better appreciate that there are only two large axonemal microtubules in the cell.

6. In the legend for Fig. 2A, was the "reversal from pushing down on coverslip" from the coverslip settling, or was additional force applied?

> No additional force was applied to the coverslip. We have clarified this in the legends of Fig. 2 and Fig. EV2 in the revised manuscript (p. 26 and p. 32).

7. When discussing Fig. 5 in the main text, it would be helpful to emphasize there was no heat shock.

> We have done so in the revised manuscript (p. 11).

8. In Fig. S5E-F, how were cell borders determined?

> Cell borders were determined by manual inspection of the DIC channel in multiple optical slices to ensure accurate delimitation. That this is the case is now indicated explicitly (p. 23).

9. There seems to be a minor typo in Fig. 6A, as there are 2 labels for "DMSO I"

> Thanks for spotting this typo, which has been corrected.

10. Chloroquine is widely used to inhibit autophagy in other systems, so the authors may wish to try that here to strengthen their autophagy claims.

> Thank you for making this interesting suggestion, which we have implemented. Importantly, we found that Hydroxychloroquine prevents or delays the turnover of tubulin-containing elements, as did Bafilomycin A1, but not that of centrioles. This result strengthens the notion that axonemal fragments are eliminated via autophagosome-mediated fusion with lysosome. Why is the elimination of centrioles not affected by Hydroxychloroquine? Suggestively, some forms of autophagy are not inhibited by Hydroxychloroquine, including LC3-associated autophagy. Therefore, it is conceivable that centrioles, but not axonemal fragment, rely on such a Hydroxychloroquine-insensitive autophagy pathway for elimination. These new findings are reported in panels EV4B and discussed in the revised manuscript (p. 13, p. 10 and p. 16).

11. In the methods, it is stated that SPY-650 tubulin labeling was used for live imaging, but I cannot find this experiment, and it would be amazing to see the experiment shown in Fig. 2A (or any of the timelapse imaging) with this probe!

> The live imaging experiments (reported in Fig. 5F-I, as well as Movies S2 and S3) indeed used SPY-650 tubulin to mark centrioles, as indicated in the corresponding legends.

Reviewer #3 (Significance (Required)):

The strength of this work is that it uncovers basic cell biology of a non-model system, which has a unique lifestyle well-suited to studies about microtubule regulation. The manuscript covers many aspects of the de-flagellation process, and presents a clear, logical model for the disassembly and degradation of axonemes and centrioles. Very little literature exists on this topic, so this is a useful publication that provides important groundwork for future studies on *Naegleria* microtubules. Limitations include that some conclusions are based on cells treated with an inhibitor of actin networks, which may impact the results related to autophagy. This work will be of broad interest to those studying non-model systems, microtubule turnover, or autophagy.

Dear Pierre,

Thank you for submitting your revised manuscript. It has now been seen by two of the original referees.

As you can see, the referees find that the study is significantly improved during revision and recommends publication. However, I need you to address the points below before I can accept the manuscript.

- Please address the remaining minor concerns of referee #2.
- Please provide 3-5 keywords for your study. These will be visible in the html version of the paper and on PubMed and will help increase the discoverability of your work.
- 'Competing interests' section needs to be renamed as "Disclosure Statement and Competing Interests". The following statement needs to be included: Prof. Pierre Gönczy is a member of the Advisory Editorial Board of EMBO Reports. This has no bearing on the editorial consideration of this article for publication."
- We note the following name discrepancy: Mary-Claude Croisier-Coeytau in the manuscript vs. Marie Croisier in eJP.
- Corresponding author needs to be clearly marked on the title page of the manuscript, and email address is necessary.
- We note the phrase "(data) not shown" on page 32, which is not allowed as per journal policy.
- We note that Figure EV6 is currently not called out in the text.
- We note that there is a legend for Figure S1 in the manuscript file and the figure is cited, but the figure is missing.
- As per the movies, the legends need to be removed from the manuscript file and provided as a readme.txt file and then each should be zipped together with its corresponding video. The correct source file names, titles and manuscript callouts should be Movie EV1-EV3.
- Please remove the instructions and the unused sections from the Reagents & Tools table.
- Please submit source data as per the email from our source data coordinator Dr. Hannah Sonntag dated 09.09.2024.
- Materials and Methods section should be renamed as Methods.
- Extended View Figures should be renamed as Expanded View Figure Legends.
- The manuscript sections should be in the following order: Title page - Abstract & Keywords - Introduction - Results - Discussion - Methods - Data Availability - Acknowledgments - Disclosure Statement & Competing Interests - References - Figure Legends - (Main Tables with legends) - Expanded View Figure Legends.
- Our production/data editors have asked you to clarify several points in the figure legends:
 - o Please note that the exact p values are not provided in the legends of figures 4g; 5c-e; 6g; EV 1c-d; EV 4a; EV 5b.
 - o Please note that information related to n is missing in the legend of figure 5d.
 - o Although 'n' is provided, please describe the nature of entity for 'n' in the legend of figure EV 5b.
 - o Please note that the scale bar is missing for figures 5a; 6c, e; EV 4d.
 - o Please note that scale bar and its definition are missing for figures EV 5e-f.
 - o Please note that the white arrowheads are not defined in the legend of figure EV 4d-e. This needs to be rectified.
- Papers published in EMBO Reports include a 'synopsis' and 'bullet points' to further enhance discoverability. Both are displayed on the html version of the paper and are freely accessible to all readers. The synopsis includes a short standfirst summarizing the study in 1 or 2 sentences (max 35 words) that summarize the paper and are provided by the authors and streamlined by the handling editor. I would therefore ask you to include your synopsis blurb and 3-5 bullet points listing the key experimental findings.
- In addition, please provide an image for the synopsis. This image should provide a rapid overview of the question addressed in the study but still needs to be kept fairly modest since the image size cannot exceed 550 (width) x 300-600 (height) pixels.

Thank you again for giving us to consider your manuscript for EMBO Reports, I look forward to your minor revision.

Kind regards,

Deniz

--

Deniz Senyilmaz Tiebe, PhD
Senior Scientific Editor
EMBO Reports

Referee #1:

The authors have adequately dealt with my concerns and comments. I am now happy for this manuscript to be published.

Referee #2:

I thank the authors for seriously considering my feedback and completing several additional experiments. I am now satisfied with nearly all of my concerns. However, I still have two minor points, and I have identified a few typos/very minor issues the authors may wish to correct.

Minor concerns:

1. K48 and K63 ubiquitination: Thank you for including the images in Figure S1- this image is quite convincing for K63 Ubiquitination colocalizing with centrioles! However, the images shown in 6C-D are not convincing. There is PolyUbi K48/K63 signal everywhere, so anywhere that is centrin-positive will also show PolyUbi signal. Some form of quantitative analysis of colocalization (e.g. a correlation coefficient or some other metric, see DOI: 10.1152/ajpcell.00462.2010) is needed to support statements about K48 or K63 ubiquitination preceding centriole degradation, given that the examples shown in the main figure do not show obvious enrichment at the centriole.
2. Statistical testing: Apologies for my comment about t-tests being unclear. If a sample set includes more than 2 conditions, performing multiple pairwise comparisons using multiple t-tests is not appropriate (see DOI: 10.1091/mbc.E15-02-0076). An ANOVA followed by a Tukey-Kramer post-hoc test is a more appropriate statistical method.

Very minor issues the Authors may wish to address:

1. In EV4B and in the main text, there are typos "Hydroxyroquine" or "Hydroxyhloroquine" instead of Hydroxychloroquine.
2. In the introduction at the top of page 4 I believe it should be "no nucleus" not "not nucleus," and it sounds like "which" points to the nucleus, but I am sure you intend to point to the flagellum.
3. The term "basal eukaryote" has a similar implication as "lower eukaryote." Early branching (relative to animals) would be a better term.

Swiss Federal Institute of
Technology Lausanne

Pierre Gönczy
Swiss Institute for Experimental
Cancer Research (ISREC)
School of Life Sciences
Swiss Federal Institute of
Technology Lausanne (EPFL)

pierre.gonczy@epfl.ch
Phone: +41 21 693 07 11
<https://gonczy-lab.epfl.ch/>

October 23rd, 2024

Re: EMBOR-2024-60279V2

Dear Deniz,

Further to your message dated October 4th, 2024, I am pleased to submit our newly revised manuscript entitled "Mechanisms of axoneme and centriole elimination in *Naegleria gruberi*" to *EMBO Reports*.

As you will see below, as well as in the revised manuscript, we have taken care of the editorial points you listed, and implemented the minor revisions suggested by Referee #2.

We hope that, with these changes, you will find our work truly ready for publication in *EMBO Reports*.

Best regards,

Pierre

- Please address the remaining minor concerns of referee #2.
We have done so (see details below).
- Please provide 3-5 keywords for your study. These will be visible in the html version of the paper and on PubMed and will help increase the discoverability of your work.
Done.
- 'Competing interests' section needs to be renamed as "Disclosure Statement and Competing Interests". The following statement needs to be included: Prof. Pierre Gönczy is a member of the Advisory Editorial Board of EMBO Reports. This has no bearing on the editorial consideration of this article for publication."
Done.
- We note the following name discrepancy: Mary-Claude Croisier-Coeytaux in the manuscript vs. Marie Croisier in eJP.
Marie Croisier is how she is referred to in publications -this has been corrected on the title page.
- Corresponding author needs to be clearly marked on the title page of the manuscript, and email address is necessary.
Done.
- We note the phrase "(data) not shown" on page 32, which is not allowed as per journal policy.
Addressed.
- We note that Figure EV6 is currently not called out in the text.
This used to be S1, we now changed the name to EV6.
- We note that there is a legend for Figure S1 in the manuscript file and the figure is cited, but the figure is missing.
Changed to EV6.
- As per the movies, the legends need to be removed from the manuscript file and provided as a readme.txt file and then each should be zipped together with its corresponding video. The correct source file names, titles and manuscript callouts should be Movie EV1-EV3.
Done.
- Please remove the instructions and the unused sections from the Reagents & Tools table.
Done.
- Please submit source data as per the email from our source data coordinator Dr. Hannah Sonntag dated 09.09.2024.
We will send these by email.
- Materials and Methods section should be renamed as Methods.
Done.
- Extended View Figures should be renamed as Expanded View Figure Legends.
Done.
- The manuscript sections should be in the following order: Title page - Abstract & Keywords - Introduction - Results - Discussion - Methods - Data Availability - Acknowledgments -

Disclosure Statement & Competing Interests - References - Figure Legends - (Main Tables with legends) - Expanded View Figure Legends.

Done.

- Our production/data editors have asked you to clarify several points in the figure legends:
 - o Please note that the exact p values are not provided in the legends of figures 4g; 5c-e; 6g; EV 1c-d; EV 4a; EV 5b.

These values are now provide in the respective figure legends.

- o Please note that information related to n is missing in the legend of figure 5d.

As stated in the legend, N=40 centrioles for each timepoint.

- o Although 'n' is provided, please describe the nature of entity for 'n' in the legend of figure EV 5b.

As stated in the legend, N= 6 (0.1% DMSO), 7 (25 μ M Bortezomib) and 5 (10 μ M Bafilomycin A1) fields of view from one representative experiment.

- o Please note that the scale bar is missing for figures 5a; 6c, e; EV 4d.

The scale bars are the same as in the neighboring image; we clarified that in the figure legends.

- o Please note that scale bar and its definition are missing for figures EV 5e-f.

E and F are zoom ins of D (green boxes). Thus, the scale bare is provided in D.

- o Please note that the white arrowheads are not defined in the legend of figure EV 4d-e. This needs to be rectified.

Done.

- Papers published in EMBO Reports include a 'synopsis' and 'bullet points' to further enhance discoverability. Both are displayed on the html version of the paper and are freely accessible to all readers. The synopsis includes a short standfirst summarizing the study in 1 or 2 sentences (max 35 words) that summarize the paper and are provided by the authors and streamlined by the handling editor. I would therefore ask you to include your synopsis blurb and 3-5 bullet points listing the key experimental findings.

We will send these by email.

- In addition, please provide an image for the synopsis. This image should provide a rapid overview of the question addressed in the study but still needs to be kept fairly modest since the image size cannot exceed 550 (width) x 300-600 (height) pixels.

We will send these by email.

Referee #1:

The authors have adequately dealt with my concerns and comments. I am now happy for this manuscript to be published.

Referee #2:

I thank the authors for seriously considering my feedback and completing several additional experiments. I am now satisfied with nearly all of my concerns. However, I still have two minor points, and I have identified a few typos/very minor issues the authors may wish to correct.

Minor concerns:

1. K48 and K63 ubiquitination: Thank you for including the images in Figure S1- this image is quite convincing for K63 Ubiquitination colocalizing with centrioles! However, the images shown in 6C-D are not convincing. There is PolyUbi K48/K63 signal everywhere, so anywhere that is centrin-positive will also show PolyUbi signal. Some form of quantitative analysis of colocalization (e.g. a correlation coefficient or some other metric, see DOI: 10.1152/ajpccell.00462.2010) is needed to support statements about K48 or K63 ubiquitination preceding centriole degradation, given that the examples shown in the main figure do not show obvious enrichment at the centriole.

We are not trying to make the point that (poly)ubiquitin is enriched at centrioles compared to other ubiquitinated targets in the cell, and apologize if we have generated this impression. Instead, we simply state that (poly)ubiquitination signals are present at centrioles, among other locations. Whereas the K63 signal can be discerned sufficiently well to predict centriolar location in approximately half of the cells, the K48 signal can be clearly discerned at centrioles only when the organelle is ideally positioned away from the cell body (as shown in the figure). Given the small size of the centriole marked by Centrin-2 antibodies, pixel correlation assessment (using ImageJ "Colocalization Test") between Centrin-2 and (poly)ubiquitination is only slightly above chance in both cases (Centrin-2 vs K63: $R=0.656R$, $R(\text{random})=0.454\pm 0.071$); Centrin-2 vs K48: $R=0.709$, $R(\text{random}): R=0.551\pm 0.065$). While these numbers support colocalization on centrioles above other targets in the cell, the difference appears too marginal to be included as an persuasive argument in the manuscript.

2. Statistical testing: Apologies for my comment about t-tests being unclear. If a sample set includes more than 2 conditions, performing multiple pairwise comparisons using multiple t-tests is not appropriate (see DOI: 10.1091/mbc.E15-02-0076). An ANOVA followed by a Tukey-Kramer post-hoc test is a more appropriate statistical method.

We have done so and corrected the manuscript accordingly.

Very minor issues the Authors may wish to address:

1. In EV4B and in the main text, there are typos "Hydroxyroquine" or "Hydroxyhloroquine" instead of Hydroxychloroquine.

Thanks, we corrected these mistakes.

2. In the introduction at the top of page 4 I believe it should be "no nucleus" not "not nucleus," and it sounds like "which" points to the nucleus, but I am sure you intend to point to the flagellum.

Correct, thanks!

3. The term "basal eukaryote" has a similar implication as "lower eukaryote." Early branching (relative to animals) would be a better term.

We agree and have changed the wording accordingly.

Prof. Pierre Gönczy
Swiss Federal Institute of Technology (EPFL)
Swiss Institute for Experimental Cancer Research
Station 19
Lausanne CH-1015
Switzerland

Dear Pierre,

Thank you for submitting your revised manuscript. I have now looked at everything and all is fine. Therefore, I am very pleased to accept your manuscript for publication in EMBO Reports.

Congratulations on a nice work!

Kind regards,

Deniz
--
Deniz Senyilmaz Tiebe, PhD
Senior Scientific Editor
EMBO Reports
